# Disruption of NIPBL/Scc2 in Cornelia de Lange Syndrome provokes cohesin genome-wide redistribution with an impact in the transcriptome

Patricia Garcia [1,14,15✉], Rita Fernandez-Hernandez[1,15], Ana Cuadrado [2], Ignacio Coca[3], Antonio Gomez [4,5], Maria Maqueda [6], Ana Latorre-Pellicer [7], Beatriz Puisac [7], Feliciano J. Ramos [7], Juan Sandoval[8], Manel Esteller [9,10,11,12], Jose Luis Mosquera [6], Jairo Rodriguez[3], J. Pié[7], Ana Losada [2] & Ethel Queralt [1,13✉]

Cornelia de Lange syndrome (CdLS) is a rare disease affecting multiple organs and systems during development. Mutations in the cohesin loader, NIPBL/Scc2, were first described and are the most frequent in clinically diagnosed CdLS patients. The molecular mechanisms driving CdLS phenotypes are not understood. In addition to its canonical role in sister chromatid cohesion, cohesin is implicated in the spatial organization of the genome. Here, we investigate the transcriptome of CdLS patient-derived primary fibroblasts and observe the downregulation of genes involved in development and system skeletal organization, providing a link to the developmental alterations and limb abnormalities characteristic of CdLS patients. Genome-wide distribution studies demonstrate a global reduction of NIPBL at the NIPBL-associated high GC content regions in CdLS-derived cells. In addition, cohesin accumulates at NIPBL-occupied sites at CpG islands potentially due to reduced cohesin translocation along chromosomes, and fewer cohesin peaks colocalize with CTCF.

[1] Cell Cycle Group, Institut d'Investigacions Biomèdica de Bellvitge (IDIBELL), Av. Gran Via de L'Hospitalet 199-203, Barcelona, Spain. [2] Chromosome Dynamics Group, Molecular Oncology Programme, Spanish National Cancer Research Centre (CNIO), Madrid, Spain. [3] Research and Development Department, qGenomics Laboratory, Esplugues de Llobregat, Spain. [4] Cancer Epigenetics and Biology Program (PEBC), Bellvitge Biomedical Research Institute (IDIBELL), L'Hospitalet de Llobregat, Barcelona, Catalonia, Spain. [5] Grup de Recerca de Reumatologia, Parc Científic de Barcelona, Barcelona, Spain. [6] Bioinformatics Unit, Institut d'Investigacions Biomèdica de Bellvitge (IDIBELL), Av. Gran Via de L'Hospitalet 199-203, Barcelona, Spain. [7] Unit of Clinical Genetics and Functional Genomics, Department of Pharmacology-Physiology and Paediatrics, School of Medicine, University of Zaragoza, CIBERER-GCV02 and IISAragon, Zaragoza, Spain. [8] Biomarkers and Precision Medicine Unit (UByMP) and Epigenomics Core Facility, Health Research Institute La Fe (IISLaFe), Valencia, Spain. [9] Josep Carreras Leukaemia Research Institute (IJC), Barcelona, Catalonia, Spain. [10] Centro de Investigación Biomédica en Red Cáncer (CIBERONC), Madrid, Spain. [11] Institucio Catalana de Recerca i Estudis Avançats (ICREA), Barcelona, Catalonia, Spain. [12] Physiological Sciences Department, School of Medicine and Health Sciences, University of Barcelona, Barcelona, Catalonia, Spain. [13] Instituto de Biomedicina de Valencia (IBV-CSIC), Valencia, Spain. [14] Present address: Instituto de Biología Funcional y Genómica, CSIC/Universidad de Salamanca and Departamento de Microbiología y Genética, Universidad de Salamanca, Salamanca, Spain. [15] These authors contributed equally: Patricia Garcia, Rita Fernandez-Hernandez. ✉email: pgarciar@idibell.cat; equeralt@idibell.cat

Cornelia de Lange syndrome (CdLS; OMIM 122470, 300590, 610759, 300882, 614701), also known as Brachmann-de Lange syndrome), is a rare, sporadic, and genetically heterogeneous autosomal- or X-linked-dominant disorder affecting multiple organs and systems during development (reviewed in refs. [1,2]). Developmental delay and intellectual disability are typically observed. The most clinically consistent and recognizable findings in CdLS patients are the facial features, limb abnormalities, small stature, and neurodevelopmental anomalies. From a molecular point of view, pathogenic variants in *SMC1A*, *SMC3*, *RAD21* (cohesin core subunits), *NIPBL* (cohesin loader), and *HDAC8* (cohesin deacetylase) can be identified in individuals with clinically diagnosed CdLS[3–12]. About 65% of CdLS cases have a heterozygous mutation in *NIPBL*, while mutations in the other four genes account for a further 11% of patients[13,14]. The molecular mechanisms of CdLS are not well understood, although the patient phenotypes suggest that the function of cohesin and NIPBL in chromatin structure becomes deregulated, thereby affecting gene transcription. Mutations in genes encoding chromatin regulators such as *ANKRD11*[15,16], *KMT2A*[17], *AFF4*[18], and *BRD4*[19,20] are also associated with a CdLS-like disorder.

Cohesin is an evolutionarily conserved ring-shaped protein complex (reviewed in refs. [21,22]) that encircles pairs of replicated chromosomes, known as sister chromatids, allowing recognition of the sister chromatids for segregation during both mitosis and meiosis[23,24]. Sister chromatid cohesion mediated by cohesin is essential for faithful chromosome segregation during cell divisions[25–27]. Cohesin is also important for the repair of DNA double-strand breaks through homologous recombination by ensuring proximity of sister chromatids, and it helps regulate chromatin structure, forming tissue and developmental specific chromatin structures that serve to mediate long-range interactions[28–33].

The core of the cohesin complex is a ring-shaped heterotrimer formed by evolutionarily conserved subunits: two SMC (structural maintenance of chromosomes) proteins SMC1A and SMC3 and the kleisin protein RAD21 (also called MDC1 or SCC1)[23,24,34]. SMC proteins are long polypeptides that dimerize and are connected by the kleisin RAD21 subunit at the ATPase heads, forming a tripartite ring-like structure. Cohesin association with chromatin is regulated by the HAWKs proteins (HEAT repeat containing proteins Associated With Kleisins) whose functions are to open, close, or stabilize the cohesin ring: STAG1/STAG2 (also called SA1/SA2), PDS5 (PDS5A or PDS5B in vertebrates) and NIPBL (also called SCC2)[35,36]. Cohesin loading to chromatin depends on the complex formed by NIPBL and MAU2 (also called SCC4) proteins[37], which facilitates topological cohesin loading in vitro by stimulating cohesin's ATPase activity[38,39]. Cohesin is loaded at specific chromosomal locations, including centromeres and promoters of some highly transcribed genes[40–43], and is then translocated[44–46] to its final retention sites, which are predominantly positions occupied by CCCTC-binding factor (CTCF) in the case of human chromosomes[45,47].

DNA is organized in loops and topologically associated domains (TADs) where enhancers are placed in close proximity to promoters[48], contributing to chromatin structure, gene regulation, and recombination[49,50]. CTCF and cohesin co-localize extensively in the genome[51] and both proteins are enriched at TAD boundaries[45,52–60]. TADs, in addition to CTCF-occupied regions, tend to be conserved through cell types and evolution[61]. Cohesin accumulates at transcription start sites (TSSs) of active genes in the absence of CTCF[45], while loss of cohesin complex abolishes TAD formation and associated loops[45,57], impairing enhancer-promoter communication.

NIPBL also plays a role in translocating cohesin along chromatin fibers[62,63] and an interaction between NIPBL and cohesin

after loading was consistently shown[64], suggesting that NIPBL also regulates cohesin after loading. In addition, a function for NIPBL in the formation of TADs and loops extrusion was suggested[65] and DNA loop extrusion by the human cohesin/NIPBL holoenzyme in vitro has recently been demonstrated[66–68]. Less NIPBL or cohesin and/or reduced mobility of cohesin could underlie CdLS pathogenesis. Here, we investigate the transcriptome of CdLS patient-derived primary fibroblasts and observe that genes associated with development and system skeletal organization are downregulated, providing a link to the developmental alterations and limb abnormalities characteristics of CdLS patients. Analysis of genome-wide distribution of NIPBL and cohesin in the same cells show reduced association of NIPBL and redistribution of cohesin, which accumulates at NIPBL-occupied sites where it is loaded and is reduced at cohesin-CTCF sites. We speculate that *NIPBL* mutations causing CdLS affect the loop extrusion activity of the cohesin/NIPBL holoenzyme, altering chromatin contacts that are required for proper expression of developmental genes. A reduction in 3D chromatin interactions is observed in CdLS-derived cells using an in silico intra-TAD loops prediction tool and validated by 3C-qPCR.

## Results

**Cohesin integrity is not affected in CdLS patient-derived fibroblasts.** The integrity of the cohesin complex has frequently been studied in cycling cells where cohesion is needed. For this reason, we wondered whether the cohesin subunits and the cohesin regulatory proteins are also important for non-diving cells. To investigate this, we examined the cohesin protein levels in cycling and in non-dividing quiescent cells (Fig. 1a lanes 1–2 and Supplementary Fig. 1b lanes 1–2). To obtain quiescent cells, control primary fibroblast cells were grown to confluence and nutrients were deprived (0.5% serum) in the medium. Western blots showed similar SMC1A, RAD21, STAG1, STAG2, PDS5A, and NIPBL protein levels in control cycling and quiescent cells (Fig. 1a and Supplementary Fig. 1a, b). Cohesive complexes, marked by the presence of Sororin (also called CDCA5), were only present in cycling cells (Supplementary Fig. 1b). Remarkably, the acetylation of SMC3 was similar in cycling and quiescent cells indicating that non-cohesive complexes are also acetylated. In addition, *SMC1A* and *NIPBL* genes were both widely expressed in differentiated tissues (Supplementary Fig. 1c). These results suggest that not only the core of the cohesin complex, but also their regulators, such as PDS5 which regulates cohesin dynamics and its loader NIPBL are required to ensure that cohesin fulfills its role in chromatin organization.

To study the integrity of the cohesin complex in patient samples, we isolated primary dermal fibroblast from CdLS individuals with known heterozygous mutations in the cohesin loader NIPBL. CdLS patients with missense mutations (P1–3 and P5–6), one with a one-aminoacid in-frame indel (P4, N1897del), and healthy donor of similar age as controls were used (Supplementary Table 1). Protein extracts from primary dermal fibroblast from four CdLS patients were prepared. Similar protein levels of SMC1A, RAD21, STAG1, STAG2, acetyl-SMC3, and PDS5A were observed in cycling and quiescent CdLS-derived fibroblast (Fig. 1a and Supplementary Fig. 1a, b). NIPBL protein levels were also similar to control cells in the *NIPBL*-mutated patient-derived cells. In conclusion, the CdLS phenotypes do not arise from a reduction in NIPBL or cohesin subunits protein levels.

We then addressed the question of whether the pool of chromatin-bound cohesin complex is affected by the reduction in the function of the loader NIPBL in the CdLS-derived fibroblasts. To this end, we performed chromatin fractionation experiments

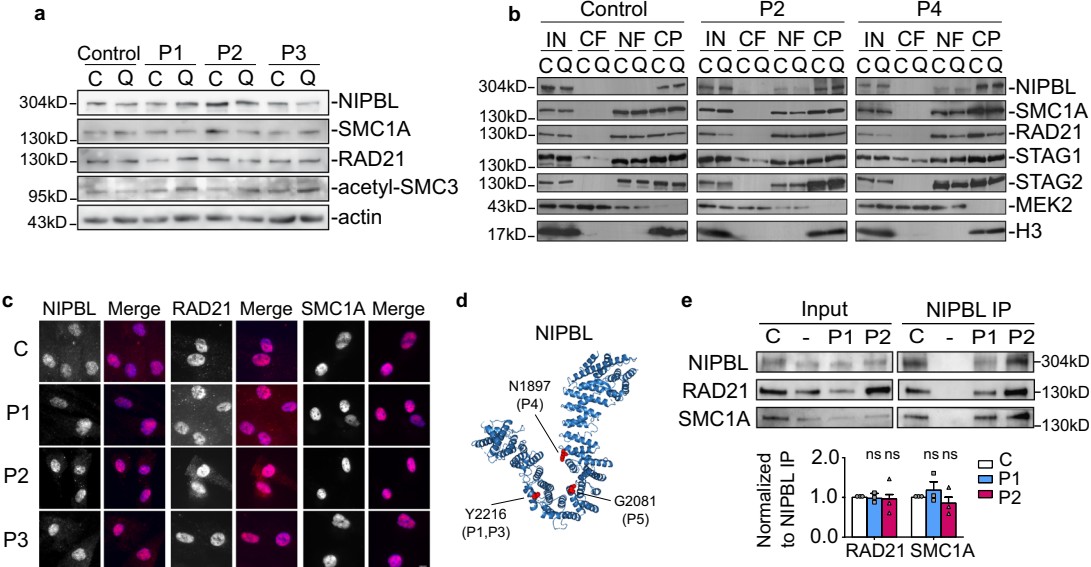

**Fig. 1 Cohesin integrity and subcellular localization are not affected in CdLS-derived fibroblasts. a** Comparison of protein levels of NIPBL, SMC1A, RAD21, and acetyl-SMC3 in primary fibroblasts derived from a control and three CdLS patients (P1–3) under cycling (C) and quiescent (Q) conditions. Actin was used as the loading control. Representative images of one experiment are shown and quantifications of three biological replicates are included in Supplementary Fig. 1a. **b** The chromatin-bound levels of NIPBL, SMC1A, RAD21, STAG1, and STAG2 proteins in two CdLS patients (P2 and P4) and a control were assessed by chromatin fractionation under cycling (C) and quiescent (Q) conditions. IN, input; CF, cytosolic fraction; NF, nuclear-soluble fraction; CP, chromatin pellet. MEK2 (cytosol) and H3 (nuclear) were used as fractionation controls. Representative blots from one biological triplicate are shown. **c** Nuclear localization of RAD21, SMC1A, and NIPBL were monitored by immunofluorescence of fibroblasts derived from a control and three CdLS patients (P1–3). Nuclei were stained using DAPI and merge signal (blue channel+ red channel) is shown. Scale bar, 10 μm. Representative images are depicted and quantifications from more than 3 biologically independent experiments are shown in Supplementary Fig. 2a. **d** Representation of the mutated residues of NIPBL in the CdLS patient-derived cells under study using Pymol software (Pymol Molecular Graphics System Version 2.0 Schrödinger, LLC) and the *Ashyba gossypii* Scc2 structure from Chao et al.[69]. Y2216 corresponds to the residue mutated in P1 (Y2216C) and P3 (Y2216S), N1897del in P4, and G2081A in P5 marked as red dots. **e** Co-immunoprecipitation experiments between NIPBL and cohesin subunits. Similar co-purification efficiencies were observed in RAD21 and SMC1A upon NIPBL immunopurification in control and two CdLS patient cells (P1 and P2). Rabbit IgG was used as negative control (marked as −) for the IP. (Lower panel) Quantification of relative protein levels of RAD21 and SMC1A normalized to immunoprecipitated-NIPBL. Means and SEMs of three independent biological replicates are shown. Two-sided unpaired student's *t*-test. Control, white; P1, blue; P2, purple.

in CdLS patient-derived fibroblasts from two CdLS patients. No significant differences were observed in the amount of NIPBL and cohesin subunits on chromatin-bound fractions (Fig. 1b, CP). Most of the NIPBL protein was found in the chromatin pellets in control and CdLS-derived fibroblasts. NIPBL was also detected at low levels in the nuclear-soluble fractions (NFs) in CdLS-derived fibroblasts, but not in control cells, suggesting that NIPBL chromatin association might be slightly altered in the former group of cells. We can conclude that bulk DNA association of the cohesin complex and NIPBL is not greatly affected in CdLS-derived cells. Nevertheless, we decided to analyze SMC1A, RAD21, and NIPBL protein localization in individual cells by immunofluorescence. No significant differences of nuclear SMC1A, RAD21, and NIPBL signals were detected in CdLS-derived cells in cycling (Fig. 1c and quantification in Supplementary Fig. 2a) and quiescent cells (Supplementary Fig. 2b, c). Altogether, we can conclude that the cohesin subunits and NIPBL are properly located at the nucleus and associated with the chromatin in CdLS-derived cells.

The C terminus hook structure of NIPBL is responsible for catalyzing cohesin loading and interacting with RAD21[69,70]. Some missense CdLS mutations were previously mapped within the NIPBL hook region[69,70]. Most of the residues mutated in the patients in our study were also located within the NIPBL hook region (Fig. 1d). To test whether mutated NIPBL proteins in CdLS-derived cells perturb the interaction with RAD21 co-immunoprecipitation experiments were performed. Similar co-purification efficiencies were observed among RAD21 and

SMC1A upon NIPBL immunopurifications (Fig. 1e), indicating that mutated NIPBL in CdLS-derived cells still interacts with cohesin and SMC1A.

**NIPBL stabilizes cohesin chromatin association.** Regulation of cohesin dynamics is essential for its proper function. To investigate the mobility of the chromatin-associated cohesin we analyzed cohesin dynamics using inverse fluorescence recovery after photobleaching (iFRAP) experiments. We transfected control and CdLS-derived cells with EGFP-RAD21[71]. The entire cell, except for a small nuclear region, was photobleached and the fluorescence redistribution in both the bleached and unbleached regions calculated (Fig. 2a, b). Recovery of fluorescence was faster in the CdLS-derived cells than in the control cells, suggesting that cohesin is less stably bound. Accordingly, recently it has been described that impairment of the CTCF and cohesin interaction renders cohesin more dynamic in iFRAP experiments, suggesting that CTCF stabilizes cohesin at cohesin-CTCF sites[72].

The fluorescence in the unbleached region decreases upon photobleaching of the adjacent regions due to diffusion of the soluble EGFP-RAD21. Therefore, the drop in mean fluorescence intensity in the unbleached region after photobleaching was used to calculate the amount of chromatin-bound RAD21[71]. Chromatin-bound cohesin becomes destabilized, as indicated by its lower level in CdLS-derived cells (Fig. 2c), which is around 55% of that in control cells. We conclude that the association of cohesin with chromatin is more dynamic in CdLS-derived cells.

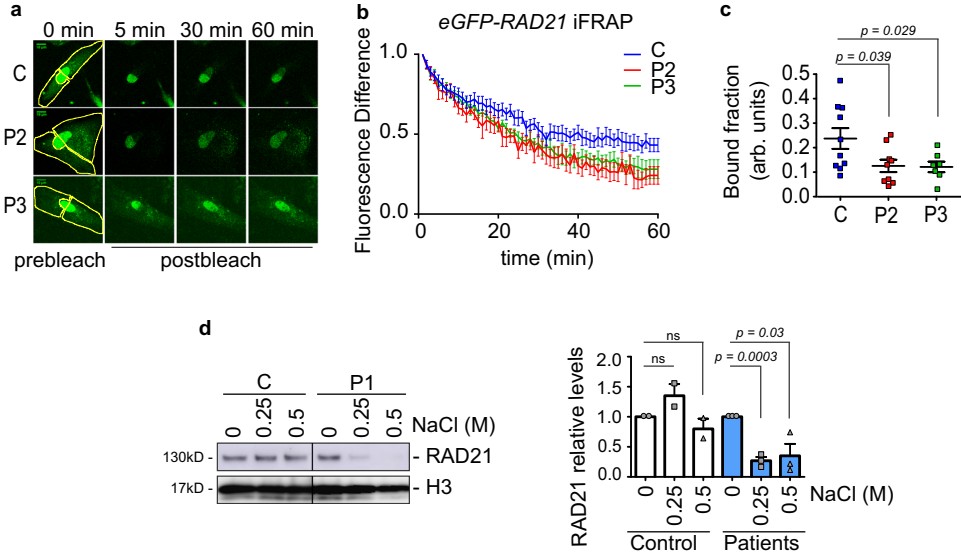

**Fig. 2 Chromatin-bound cohesin is less stable in CdLS-derived cells. a** Representatives images of an iFRAP experiment of control (C) and two CdLS patients (P2–3) cells before (0 min) and 5, 30, and 60 min after photobleaching (green channel). The entire cell, except for a small nuclear region, was photobleached (yellow line) and the fluorescence of *EGFP-RAD21* redistribution in the bleached and unbleached regions was followed by time-lapse microscopy. **b** Fluorescence recovery after bleaching was quantified as the difference between bleached and unbleached nuclear regions over 1 h in one control (C, blue) and two CdLS patients (P2, red; P3, green) and normalized to the first post-bleach frame after 2 h min. Means and SEMs are shown. Control, $n = 10$; P2, $n = 8$; P3, $n = 7$ cells examined over more than 3 biologically independent experiments. Two-sided unpaired student's *t*-test. **c** The drop in mean fluorescence intensity in the unbleached region at the first post-bleach frame was used to calculate the chromatin-bound fraction. Same cells than in (**b**) were used. Means and SEMs are shown. Two-sided unpaired student's *t*-test. **d** Chromatin fractions were prepared from cells derived from control (C) and CdLS patient (P) derived cells and washed with 0.25 and 0.5 M of NaCl containing buffer for 30 min. The amount of RAD21 bound to the chromatin was analyzed by immunoblotting. A representative experiment is shown (left panel), in which histone3 (H3) was used as a loading control. Plots of RAD21 levels in controls (white) and 3 biologically independent samples (CdLS patients P1, P2, and P3, blue) are depicted (right panel). Means and SEMs are shown. Two-sided unpaired student's *t*-test.

To confirm that cohesin is more dynamic in CdLS-derived cells than in control cells, we performed a salt extraction on chromatin fractionations. Chromatin fractions were treated with increasing amounts of NaCl (0.25 and 0.5 M) and the amount of remaining RAD21 on chromatin was determined (Fig. 2d). The amount of RAD21 associated with chromatin in the CdLS-derived cells was around 70% less than in control cells, indicating that RAD21 was more sensitive to salt concentrations in CdLS-derived cells. All these results indicate that the association of cohesin with chromatin is less stable in CdLS-derived cells suggesting that NIPBL stabilizes cohesin on DNA.

**Genes involved in development and differentiation are deregulated in CdLS-derived cells**. We next analyzed the transcriptomes of primary dermal fibroblasts from CdLS individuals. Early passages of primary dermal fibroblast from five *NIPBL*-mutated CdLS patients (P2–P6) and four control donors were used in Agilent SurePrint G3 Human Gene Expression Microarrays v2 (Supplementary Data 1). A statistical analysis (FDR < 0.25 and |logFC|>0.5) identified 3224 probesets containing 2049 differentially expressed genes (DEGs, Supplementary Data 2) (1153 upregulated and 896 downregulated), with logFC values ranging from 5.2 to −2.4 (Fig. 3a). Moreover, 506 of the 2049 (25%) genes exhibit a >1.5-fold change.

A Gene Ontology (GO) enrichment analysis terms revealed functional differences between the CdLS expression profile compared to control cells (Fig. 3b and Supplementary Table 2). Genes related to embryonic development were downregulated. In contrast, GO-defined processes related to translational initiation and mitotic processes were upregulated.

Our GO analysis showed a deregulation of genes involved in embryonic system development and differentiation (Fig. 3c),

including genes related to central nervous system development and differentiation (Fig. 3c, marked in blue). Among the genes involved in neural differentiation are the protocadherins (*PCDH*) which are known to be downregulated in cohesin-SA1 and Nipbl heterozygous mice models[73,74]. The expression changes of several protocadherins genes and *SHC3* involved in neuron development were validated by RT-qPCR (Fig. 3d).

*HOX* (Homeobox family of genes) genes encode for master transcription factors responsible for patterning the anterior–posterior body axis during embryonic development. Several *HOX* genes proved to be deregulated in our transcriptome analysis (Supplementary Data 2 and Fig. 3c marked in red). Validation by RT-qPCR confirmed the upregulation of *IRX2* involved in embryonic development and several genes in the HOXB and HOXC clusters (Fig. 3e). We also corroborated the downregulation of the genes in the HOXD cluster and the Hox-regulated factor *HAND2*. *HOXD* genes are expressed in the developing limb with a characteristic proximal-distal pattern. Deletions that remove the entire *HOXD* gene cluster or the 5′ end of the cluster have been associated with severe limb abnormalities, similar to those found in CdLS patients[75].

Gene expression studies do not distinguish among deregulated genes directly link to *NIPBL* mutations or due to indirect secondary effects. To elucidate between these two possibilities, we introduced wild-type *NIPBL* in CdLS-derived cells from Patient 3 (Supplementary Fig. 3). Nucleofection with commercially available *GFP-NIPBL* (IDN3 from Origene) resulted in 300 times overexpression of *NIPBL* mRNA (Supplementary Fig. 3a). Remarkably, re-introduction of wild-type *NIPBL* rescued the altered expression of central nervous system development and differentiation genes, such as the *PCDHB* family, *SHC3*, and *MEF2C* (Supplementary Fig. 3b). In addition, 70% of the tested genes (7/10) involved in embryonic system

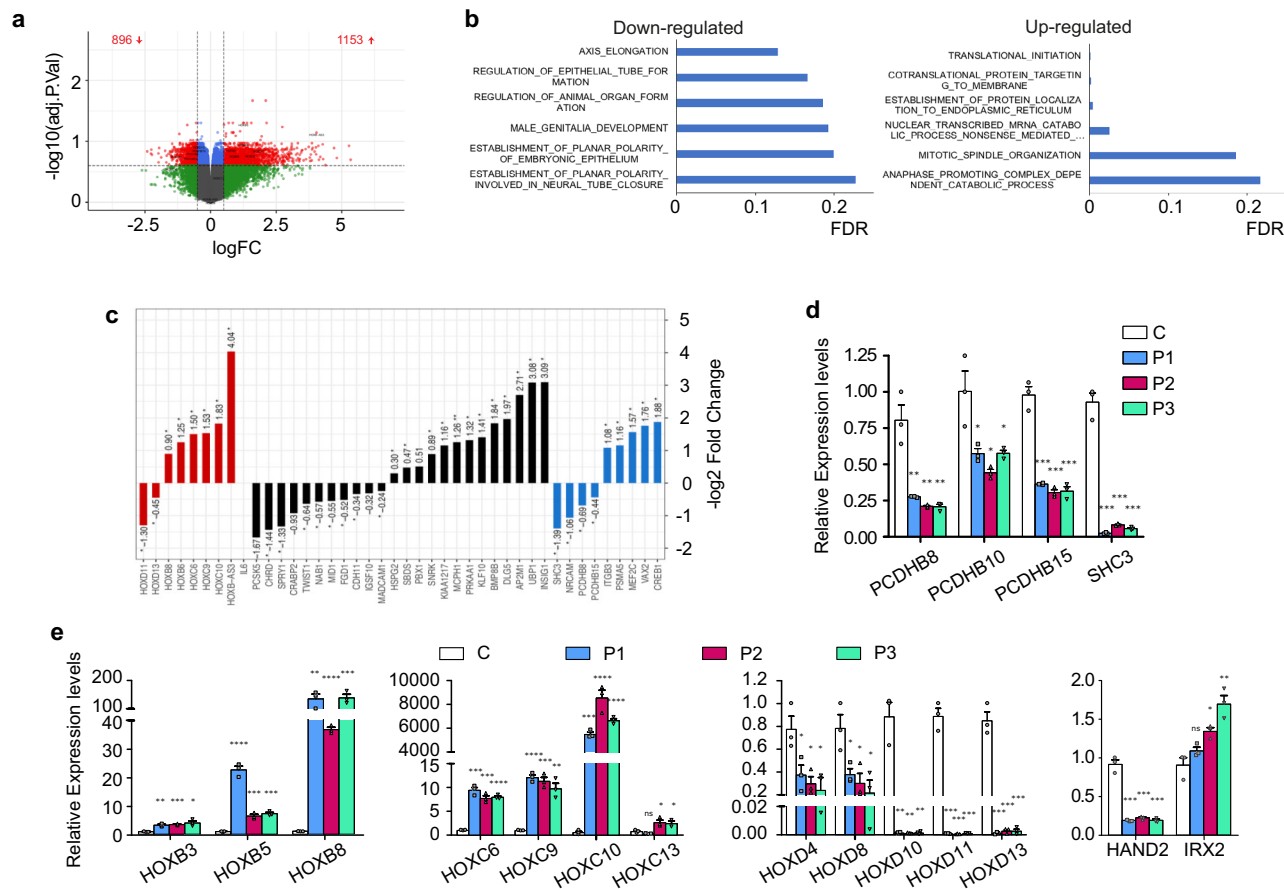

**Fig. 3 Genes involved in development and differentiation are deregulated in CdLS patient-derived cells. a** Volcano plot representation of gene expression changes between control and CdLS-derived fibroblasts. Significantly downregulated and upregulated genes are indicated in red. Green, blue, and gray show non-significant differential genes. **b** Gene ontology (GO) enrichment analyses reveal biological processes that are downregulated and upregulated in CdLS-derived fibroblasts (blue bars). **c** Representation of the log-fold changes in expression of genes related to developmental processes (black bars). Genes involved in embryonic system development and differentiation are marked in red, those involved in nervous system development are marked in blue. Two-sided unpaired student's *t*-test (\*\**p* < 0.01; \**p* < 0.05). **d, e** The expression levels of some nervous system development genes (**d**) and embryonic development genes (**e**) were analyzed in a control (C, white) and three CdLS patient-derived fibroblasts (P1, blue; P2, purple; P3, green) by reverse transcription-qPCR. The graphs show the amount of transcript of each gene relative to that in the control. Means and SEMs were calculated from biological triplicates. Two-sided unpaired student's *t*-test (\*\*\*\**p* < 0.0001; \*\*\**p* < 0.001; \*\**p* < 0.01; \**p* < 0.05).

development recovered the gene expression similar to control cells (Supplementary Fig. 3c, marked in blue). In conclusion, our results suggest that deregulation of most of the embryonic and central nervous system genes observed in CdLS-derived cells are directly linked to *NIPBL*.

Strikingly, we found the same gene expression deregulation of *PCDH* and *HOX* genes in two patients with mutation in the *SMC1A* cohesin subunit (Supplementary Fig. 4), suggesting that the transcriptome changes observed in *NIPBL*-mutated CdLS patients could be extended to patients bearing other mutations.

**Genome-wide distribution of NIPBL in CdLS-derived cells.**
Cohesin distribution on chromatin is usually a good readout for its contribution to 3D genome organization and gene regulation. To gain insight into the genome-wide distribution of cohesin and NIPBL we performed ChIP-sequencing experiments. Several commercial NIPBL antibodies were validated to select the best available antibody for ChIP-seq (Supplementary Fig. 5). NIPBL antibody (3B9) from Novus gave the best immunoprecipitation efficiency and had the expected localization in the nucleus by immunofluorescence, so we used this antibody for the ChIP-seq experiments. To be able to quantitatively compare the profiles obtained in fibroblasts from patients and healthy donors, we

carried out a quantitative spike-in ChIP-seq experiment. We identified 75,872 NIPBL sites in control cells (Supplementary Data 3). Representation of the NIPBL peaks around two genomic regions, with and without densely packed genes regions, are shown (Fig. 4a). NIPBL sites were previously reported[43] as being most frequently found at gene promoters. We detected 8372 (11%) NIPBL peaks at promoters (Fig. 4b) and considering an average of 20,000 genes in the human genome, we can conclude that almost half of the gene promoters are occupied by NIPBL. We also found 33% of NIPBL peaks to be located near enhancers (Fig. 4b). Remarkably, the most important feature is that NIPBL colocalized with DNA regions with high GC content (GC > 60%). Around 67% of NIPBL peaks appeared in high GC regions (Fig. 4b), including 38% (19,536 NIPBL peaks) in CpG islands. The human genome is estimated to contain around 30,000 CpG islands, meaning that 65% of all CpG islands are occupied by NIPBL. We found that NIPBL peaks at CpG islands coincide with promoters (39%) and with enhancers (29%) (Fig. 4c). CpG islands tend to accumulate at promoters, enhancers, and developmental gene clusters[76]. By examining the NIPBL peak distribution with respect to the chromosomal distance (Fig. 4d), we observed that many NIPBL peaks appear close together, with 15,804 NIPBL (21%) peaks located within <2000 bp.

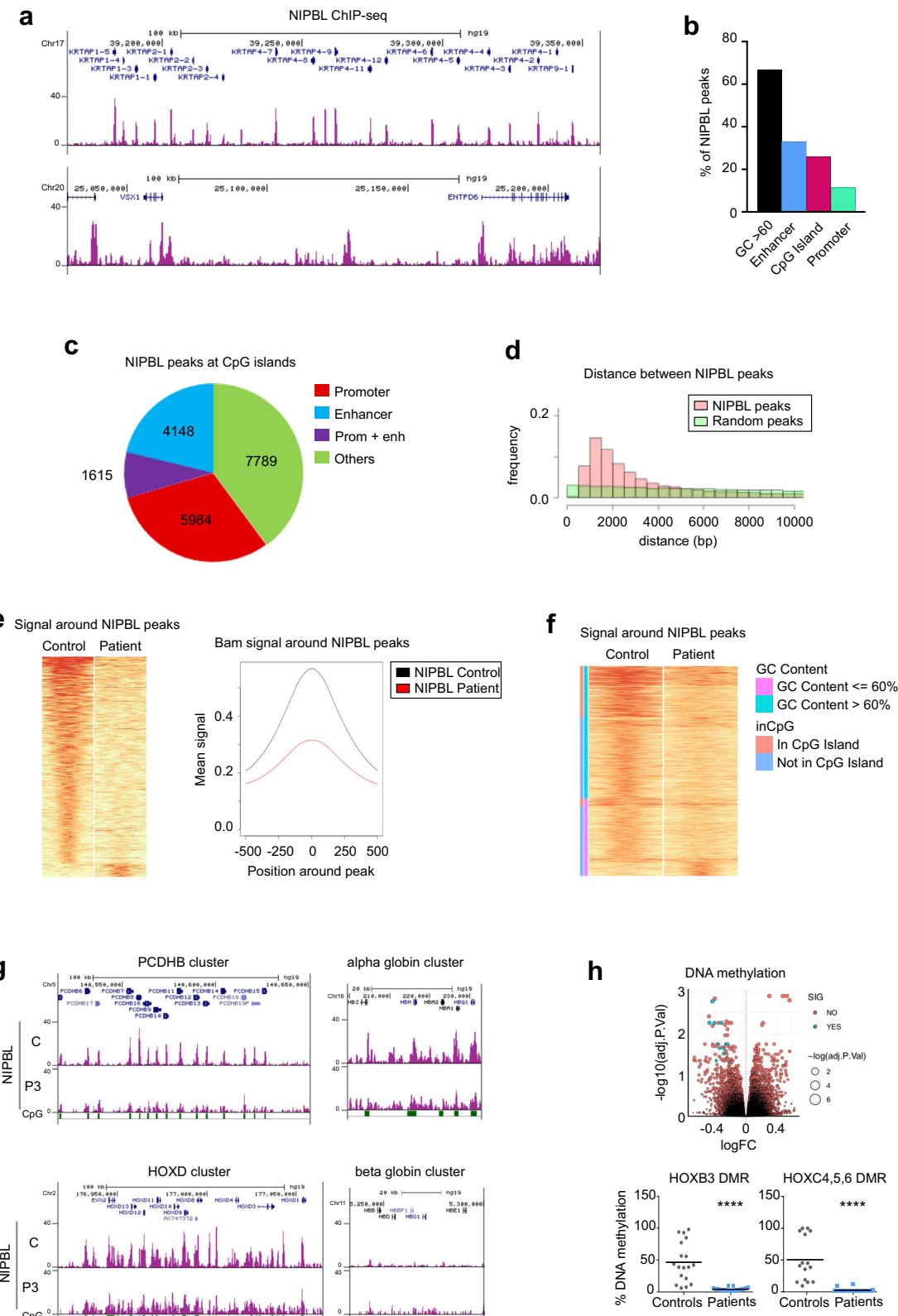

Quantitative comparison between the NIPBL ChIP-seq analyses in control and CdLS-derived fibroblasts from Patient 3 showed a great loss of NIPBL signal in the patient (Fig. 4e, f; Supplementary Data 3). We identified 20,428 NIPBL sites in CdLS-derived cells which correspond to a reduction of 73% of the called peaks compared with the control. Validation by NIPBL ChIP-qPCR was done in other three patients (P1, P2, and P4) in 5

regions including the HOXB, C, and D (EVX2 position) clusters (Supplementary Fig. 6). We conclude that mutated NIPBL in CdLS patients associates with chromatin less efficiently than it does in control cells.

From the genes differentially expressed in CdLS-derived cells (Supplementary Data 2), 40% contained NIPBL at their promoters, including *HOX* genes. We observed an aggregation

**Fig. 4 Genome-wide distribution of NIPBL in CdLS-derived cells. a** Snapshots of the browser showing representatives NIPBL ChIP-seq data in control fibroblasts in two different genomic regions. **b** NIPBL peaks distribution. Percentages of NIPBL peaks that colocalized with the indicated genomic features (black, GC > 60; blue, enhancer; purple, CpG Island; green, promoters). **c** Pie charts showing the distribution of NIPBL peaks located at CpG islands in promoters (red), enhancers (blue), or promoters + enhancers (violet) and others (green). **d** Plot representing the NIPBL peaks distribution with respect to the chromosome distance (pink) compared with random peak distribution (green). **e** Heatmap (left panel) and plot of mean read density NIPBL signals (right panel) showing a general loss of NIPBL signal around NIPBL peaks in Patient 3-derived cells (red) compared with the control (black). **f** Heatmap showing NIPBL signal distribution with respect to the DNA GC content and its presence into a CpG island and their corresponding signal in the patient (GC content <= 60%, pink; GC content > 60%, light blue; in CpG island, light red; not in CpG Island, blue). **g** Snapshots of the NIPBL peaks distribution in control (C) and CdLS Patient 3-derived cells (P3) at four gene clusters. CpG island positions are indicated in green boxes. **h** DNA methylation in CdLS-derived fibroblasts. (upper panel) Volcano plot representing the DNA methylation changes in control and CdLS-derived fibroblasts (significantly methylated samples are marked in blue and non-significant in red). Validation by bisulfite pyrosequencing was performed at six DMPs for HOXB3 (Control, n = 3, gray; Patients, n = 3, blue) and at five for HOXC4, 5, and 6 (Control, n = 3, gray; Patients, n = 4, blue) from three independent experiments (lower panel). Means were calculated for all the DMPs within a promoter and represented as DMRs. Two-sided unpaired student's t-test (****p < 0.0001).

of NIPBL peaks at the HOX and protocadherin clusters (Fig. 4g). Accumulation of NIPBL was also detected in other gene cluster regions such as alpha-globin, histone HIST1, keratin II, IRXA, and IRXB. In general, these clusters are rich in CpG islands, and chromatin-bound NIPBL is associated with the presence of CpG islands. Unlike the alpha-globin cluster, which contains CpG islands, the beta-globin cluster is an A + T rich region, does not contain CpG islands and we did not detect NIPBL chromatin association (Fig. 4g).

Since CpG dinucleotides are regulated by DNA methylation, we wondered whether DNA methylation patterns changed in our CdLS-derived cells. In genome-wide DNA methylation analysis, we found 123 differential methylation positions (DMPs) between controls and CdLS-derived cells (Supplementary Data 4). DMPs at promoters are those that can lead to a change in gene expression, and we identified only 13 of these (Fig. 4h upper panel and Supplementary Data 4) which included four DMPs at promoters of *HOX* genes. Pyrosequencing validation of the DNA methylation changes in three controls and four patients confirmed the decrease in the differential DNA methylation regions (DMR) at HOXB3 and HOXC4,5,6 promoters in CdLS-derived cells (Fig. 4h lower panel). The decrease in DNA methylation is associated with an increase in gene expression, as shown in (Fig. 3e). In conclusion, the overall differences in gene expression in CdLS-derived cells are not due to the defective DNA methylation of the promoters of the corresponding genes, consistent with the fact that most CpG islands at promoters are demethylated. However, in the *HOXB-C* genes, DNA methylation may also contribute to the differential expression detected.

**Genome-wide distribution of cohesin in CdLS patient-derived fibroblasts**. Since NIPBL is the cohesin loader we performed SMC1A ChIP-sequencing experiments with a control and two CdLS-derived fibroblasts (P3 and P4). We identified 22,957 SMC1A peaks in the control and 26,190 in the P3 CdLS-derived cells (Supplementary Data 5). Our SMC1A ChIP-seq data overlap closely with RAD21 ChIP-seq data from Encode (Fig. 5a, b). Venn diagrams showed a high degree (91% of sites) of overlap of SMC1A and RAD21 sites, as expected (Fig. 5b), while the overlap with NIPBL was reduced to 9.7% sites. Consistently, NIPBL peaks overlap only slightly with RAD21 and CTCF from Encode (21% with RAD21 and 26% with CTCF).

Although SMC1A distribution appears similar in control and CdLS-derived cells, a careful analysis using the MaNorm software identified 1732 differential peaks between the control and P3 (Supplementary Data 6), of which 1082 are gained and 649 are lost (Fig. 5c). Importantly, sites where we identified SMC1A gains are characterized by the presence of NIPBL peaks, with 69% of all gained SMC1A peaks being co-occupied by NIPBL (Fig. 5c). Moreover, genomic locations where we detected SMC1A gains are also characterized by a low or absent

CTCF occupancy, a moderate DNase accessibility, and low MNase signal (Fig. 5d, left panel). Interestingly, genomic locations where we identified SMC1A peak gains show a decrease in the NIPBL occupancy levels in the CdLS-derived cells. By contrast, the SMC1A sites that are lost in the patients are characterized by a weak NIPBL signal, a strong high CTCF signal, and two well-defined MNase peaks flanking the CTCF peak, which is indicative of the presence of defined cohesin-CTCFs sites (Fig. 5d, right panel). Similar results were obtained in the SMC1A ChIP-seq for a second patient, Patient 4 (Supplementary Fig. 7, Supplementary Data 7). An increase of the SMC1A signal at CpG islands colocalizing with NIPBL was observed, while reduction of the SMC1A signal mainly occurred at cohesin-CTCF sites (Supplementary Data 8, Supplementary Fig. 7c–e). Taken together, our results clearly indicate that the chromatin regions, where SMC1A is gained and lost, are structurally very different: while SMC1A tends to be gained at NIPBL-occupied, CTCF-free, moderately accessible regions; SMC1A tends to be lost at regions where it is usually found: CTCF-occupied and NIPBL-free regions.

ChIP-seq shows that NIPBL is mostly found in high GC content regions, with a subpopulation of peaks corresponding to CpG islands. Thus, we wondered whether the differential SMC1A peaks colocalize with NIPBL at CpG islands. This is indeed the case, since of all SMC1A peaks that overlap with NIPBL, 85% corresponded to CpG islands (Fig. 5e).

Since CpG islands are commonly found at promoters and enhancers, we interrogated the genome-wide distribution of the SMC1A differential peaks (Fig. 5f). With regards to gained peaks, 61% of the SMC1A peaks corresponded to CpG islands; and 31 and 33% overlapped with promoters and enhancers, respectively. As expected, most of the promoter sites contained CpG islands, while only half of the enhancers did so. By contrast, 55% of the SMC1A lost peaks colocalized with CTCF. These results confirmed that SMC1A gained peaks in the CdLS-derived cells correspond mostly to NIPBL-occupied sites at CpG islands, while peaks are lost at CTCF overlapping sites.

An increase in the SMC1A signal at NIPBL-occupied CpG island was validated at 10 positions in two patients (Fig. 6a). Next, we studied the gene ontology of the SMC1A differential peaks and terms related to development and transcription were greatly enriched (Fig. 6b, Supplementary Data 9), including *HOX* genes. We validated the results by SMC1A ChIP-seq qPCR and confirmed that SMC1A is enriched at cohesin-CTCF sites in HOX and PCDH clusters in control and CdLS-derived cells (Fig. 6c). SMC1A DNA binding was reduced in CdLS-derived cells at four out of the seven HOX and three out of the four PCDH sites tested. In conclusion, the differential SMC1A peaks appeared close to genes related to development and transcription.

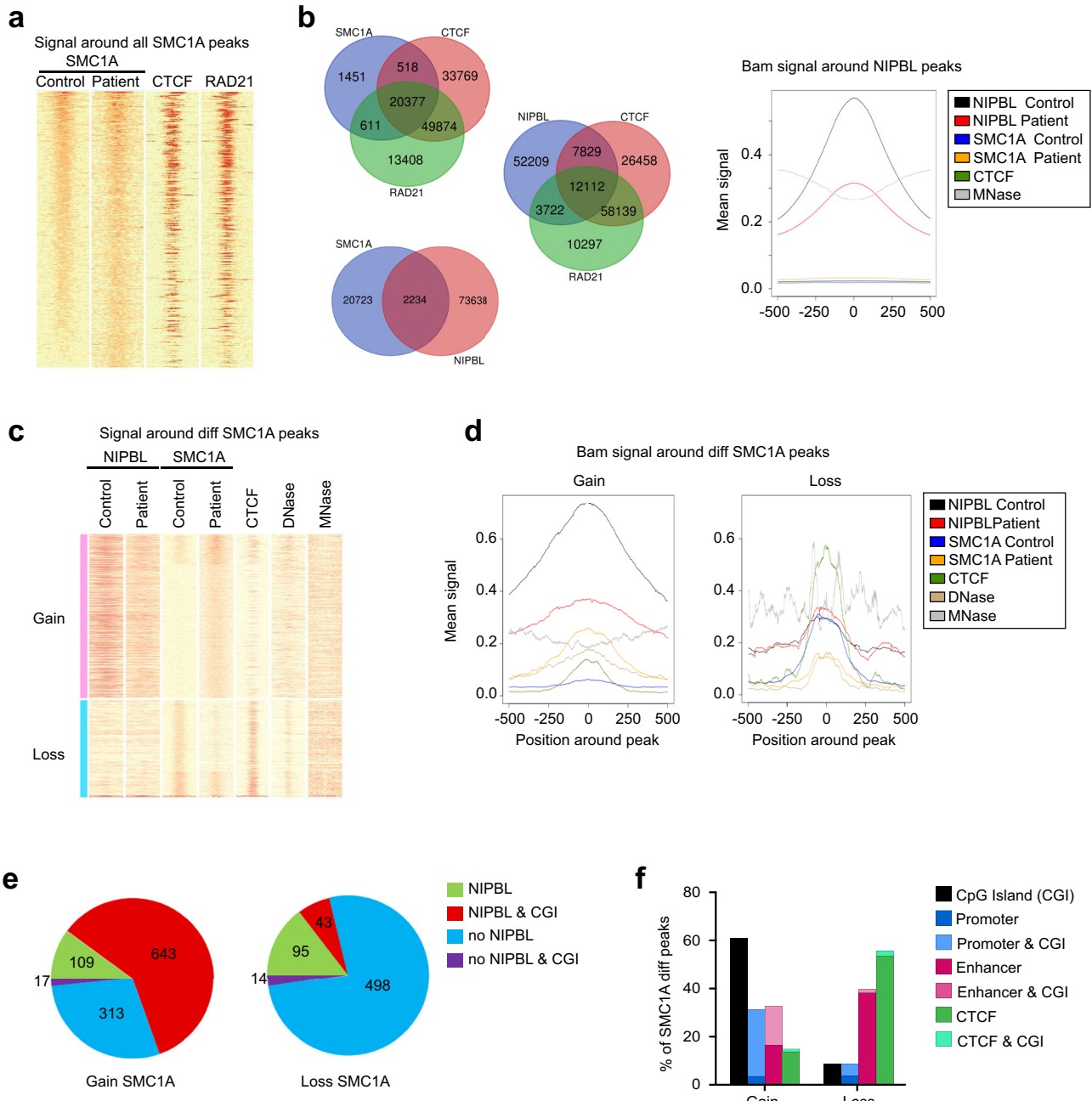

**Fig. 5 Genome-wide distribution of SMC1A in CdLS-derived cells. a** Heatmap showing the signal around all SMC1A peaks in control and CdLS patient 3-derived cells and its colocalization with CTCF and RAD21. **b** (left) Venn diagrams showing the overlap of genomic positions of SMC1A (dark blue) with CTCF (light red) and RAD21 (green) (up), SMC1A (dark blue) with NIPBL (light red) (bottom) or NIPBL (dark blue) with CTCF (light blue) and RAD21 (green) (right). **b** (right) Bam graph showing the mean signal of NIPBL (Control, black; Patient, red), SMC1A (Control, blue; Patient, orange), CTCF (green), and MNase (gray) around NIPBL peaks in control and CdLS patient 3-derived cells. **c** Heatmap comparing the signal in control and CdLS patient 3-derived cells around differential SMC1A peaks. Signals of SMC1A, NIPBL, CTCF, DNase, and MNase genomic distribution are depicted. The heatmap is divided into two categories, SMC1A peaks gained (pink) or lost (light blue) in the patient compared with the control. **d** Bam graphs showing mean signal of the indicated proteins around gained (left) and lost (right) SMC1A genomic positions. NIPBL (Control, black; Patient, red), SMC1A (Control, blue; Patient, orange), CTCF (green), DNase (brown), and MNase (gray). **e** Pie charts showing the overlap of gained (left) and lost (right) SMC1A peaks with NIPBL and CpG islands (CGI). NIPBL, green; NIPBL&CGI, red; no NIPBL, blue; no NIPBL&CGI, violet. **f** Plots of percentages of SMC1A gained and lost peaks overlapping with CpG islands (CGI, black), promoters (blue), enhancers (pink), and CTCF (green).

**Reduction of chromatin contacts in CdLS patient-derived fibroblasts.** Using a computational method previously described[77] we predicted 3D-looped intra-TAD structures based on CTCF motif orientation (1D) and CTCF and cohesin ChIP-seq binding strength data (2D). To validate the method, we compared the in silico predicted intra-TAD loops using the data

from our control sample SMC1A ChIP-seq with published Hi-C data for a fibroblast cell line IMR90[78]. We identified 7268 experimental loops for IMR90 with a median loop size of 239 kb (IQR 290 kb) containing a CTCF motif within both their boundaries (Supplementary Data 10). On the other hand, we predicted 17,461 TADs in silico from the control sample with

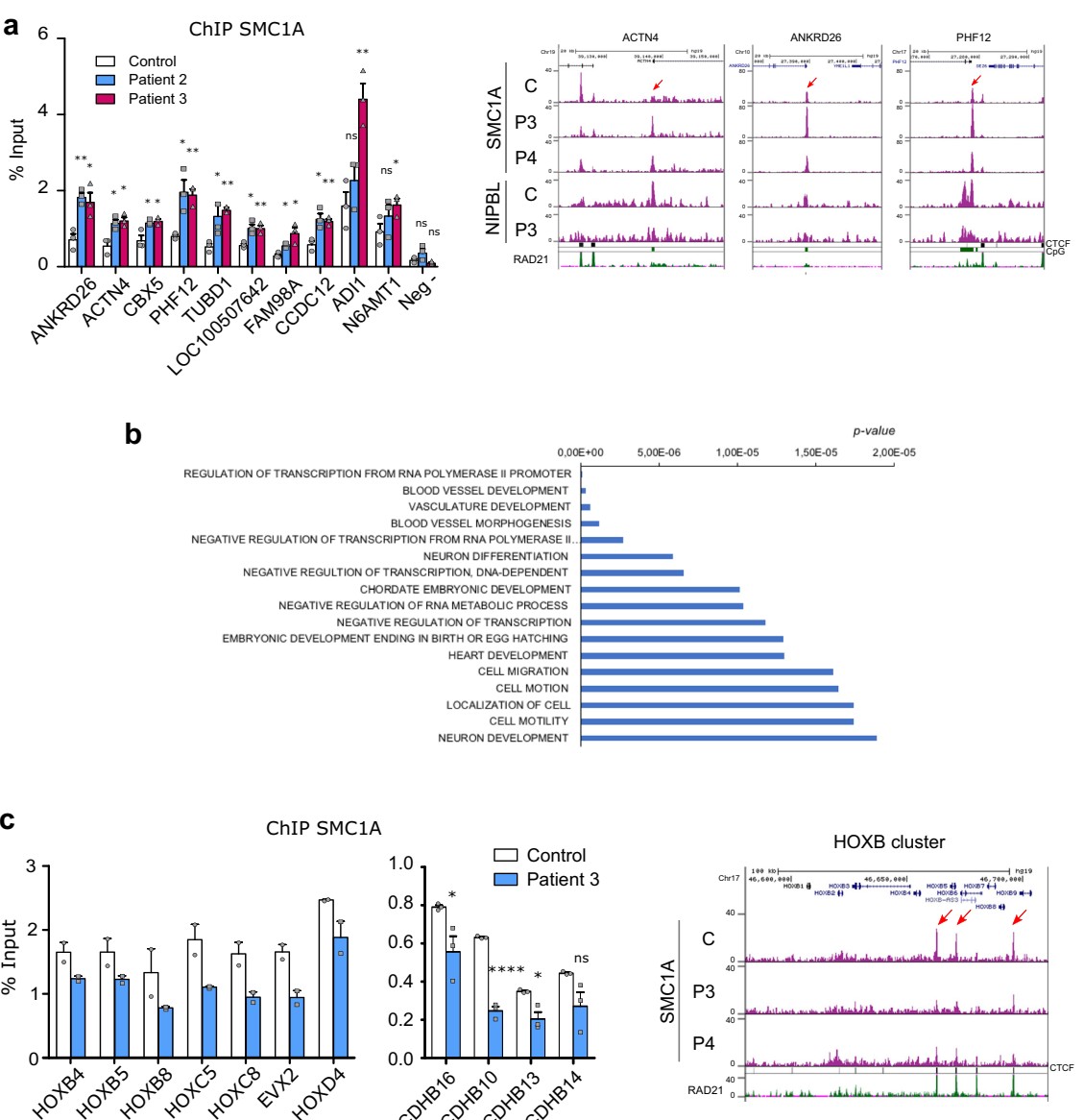

**Fig. 6 Validation of the differential SMC1A peaks by ChIP-qPCR experiments. a** Cells from a control (white) and two CdLS patients (P2, blue; P3 purple) were used in ChIP-qPCR experiments to validate the SMC1A ChIP-Seq data using primers at the indicated genes. An intergenic region was used as a negative control (Neg −). Means and SEMs were calculated from biological triplicates (left). Two-sided unpaired student's *t*-test (**p < 0.01; *p < 0.05). Snapshots of SMC1A and NIPBL peak distributions in control (C) and CdLS patient cells (P3–4) at three validated genome regions (right). The red arrows indicate the validated SMC1A peaks. **b** Gene ontology (GO) analysis reveals enriched biological processes in the SMC1A differential genomic positions in CdLS-derived fibroblasts. **c** ChIP-qPCR experiments were performed to validate the SMC1A ChIP-Seq data at three regions of HOXB, two of HOXC, two of HOXD, and four regions of the PCDHB clusters using cells from a control (white) and a CdLS patient (blue). Means and SEMs were calculated from biological duplicates (HOX) and triplicates (PCDHB). Two-sided unpaired student's *t*-test (****p < 0.0001; *p < 0.05) (left). Snapshots of SMC1A peak distributions in control (C) and CdLS patient cells (P3–4) for the HOXB cluster (right). Red arrows indicate three validated SMC1A peaks within the HOXB cluster.

229.7 kb median size (IQR 339.1 kb) (Supplementary Data 10). Then, we quantified the in silico and experimental TADs that reciprocally overlapped (complete match) or partially matched (Fig. 7a, see "Methods" for more details). Most of the experimental loops (around 85%) completely (38.9%) or partially (52.9%) matched with the predicted intra-TADs (Supplementary Fig. 8a). Among them, 65.6% of loops overlap more than 75% (38.9% complete match and 26.5% of partial match) (Supplementary Fig. 8a, b). Partially matching predicted intra-TAD loops in the control sample were mainly located inside an experimental loop (15.2%) or included one (27.8%); while the remaining 56.9%

correspond to predicted intra-TADs that are displaced upstream or downstream (Supplementary Fig. 8c). These results suggest that the intra-TAD prediction tool can recapitulate most of the experimental TADs detected in Hi-C data, validating the use of the computational tool for prediction of intra-TAD loops.

Next, we used the computational tool for prediction of intra-TAD loops using the SMC1A ChIP-seq data for CdLS Patient 3 and Patient 4 and compared it with the control dataset (Fig. 7). A total of 17,766 intra-TADs were predicted for the CdLS patients. We found 25.3% intra-TAD loops that are different between the control and the patient samples: including 16.3% of unmatched

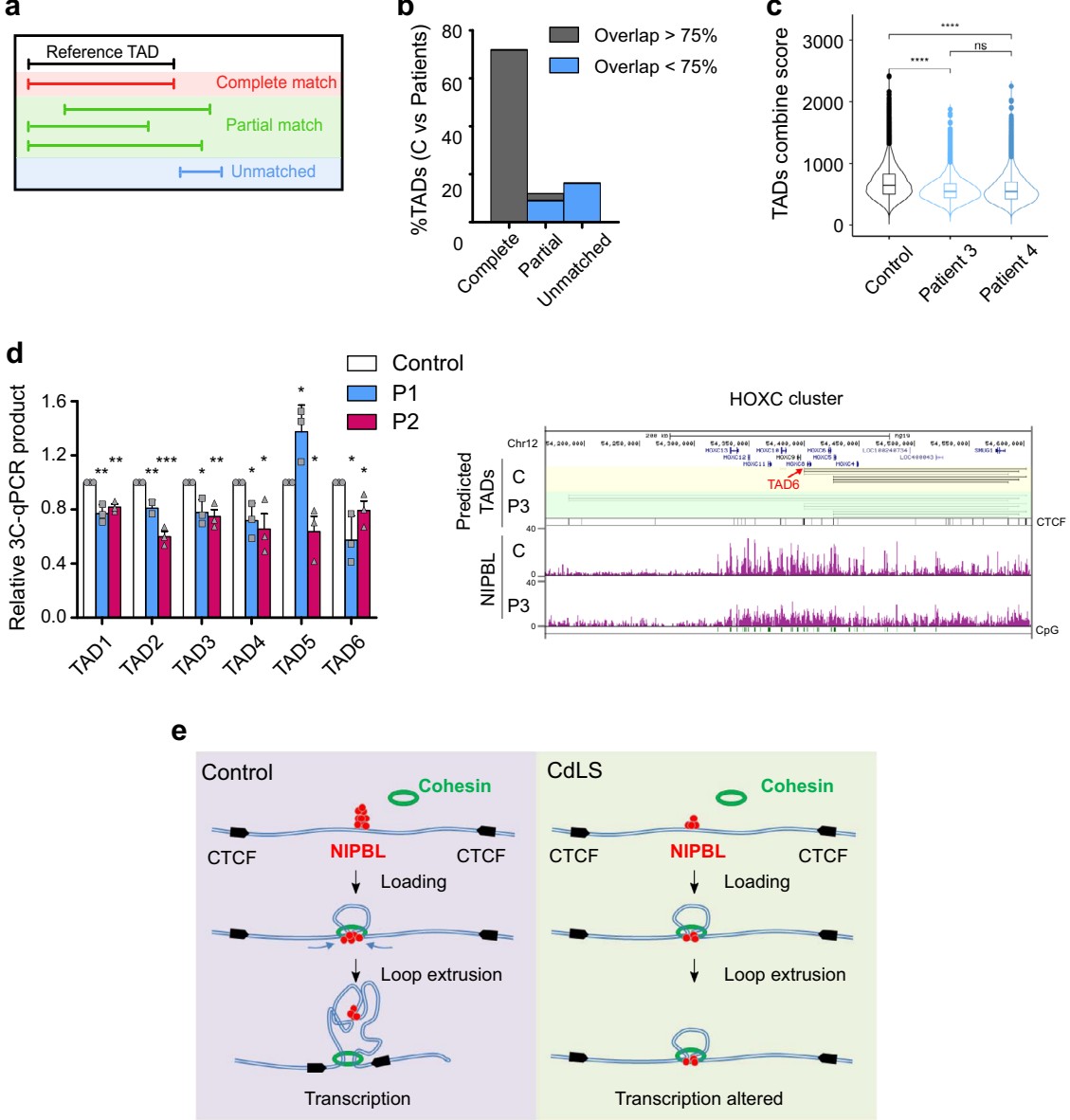

**Fig. 7 Reduction of chromatin contacts in CdLS patient-derived fibroblasts. a** Cartoon representing the complete (red), partial (green), and unmatched (blue) intra-TADs (see "Methods" for more details). **b** Bar plot showing the percentage of the predicted intra-TAD loops from CdLS patients completely (71.9%), partially (11.8%), or unmatching (16.3%) with the control intra-TADs. TADs with an overlap of more than 75% are considered unchanged (gray) between the control and the patients. TADs with an overlap lower than 75% are considered different between the control and the patients (blue). **c** Boxplots and violin plots representing the intra-TAD combined scores predicted for control (black), patient 3 (light blue), and patient 4 (steel blue). Boxplots: center line, median; box limits, first and third quartiles; whiskers, 1.5× interquartile; points, outliers. Violin plots show the corresponding probability density function. Two-sided Wilcoxon test (****$p < 0.0001$). **d** Validation of the intra-TAD loops prediction by 3C-qPCR experiments from a control and two CdLS patients from biological triplicates. Means and SEMs are shown. Two-sided unpaired student's $t$-test (***$p < 0.001$; **$p < 0.01$; *$p < 0.05$). Snapshot of a representative intra-TAD loops prediction is depicted (right). **e** A model for the differential NIPBL and cohesin genomic distribution in CdLS patient-derived cells. Under normal conditions NIPBL (red) is enriched in regions of high GC content, including promoters, enhancers, and gene clusters. Cohesin (green) is recruited at the NIPBL-occupied positions and reaches their final cohesin-CTCF (black) sites by DNA translocation and loop extrusion formation (left). NIPBL chromatin association is greatly reduced in CdLS-derived cells. Defective NIPBL function might reduce cohesin DNA translocation and/or loop extrusion kinetics. Cohesin peaks increase at the NIPBL-occupied positions and reduce at the cohesin-CTCF sites (right).

intra-TAD loops and 9% of partial intra-TAD loops with less than 75% overlap (Fig. 7b and Supplementary Data 11). In addition, control intra-TAD scores were significantly higher than both patients while between patients no significant differences were detected (Fig. 7c). In order to validate the data obtained with the intra-TAD prediction tool, we performed 3C-qPCR experiments in control and CdLS-derived cells from Patient 1 and

Patient 2. A 30% reduction in the 3C contacts was observed in the CdLS-derived cells (Fig. 7d). This suggests that the intra-TAD loops are weaker or loosen in the CdLS patients pointing out to a less constrain chromatin organization. Alternatively, a higher single-cell variability of intra-TAD loop formation in CdLS-derived cells, will also results in reduced intra-TAD scores in the whole-cell population.

## Discussion

CdLS is a cohesinopathy, also considered recently a transcriptomopathy[17]. Our data suggest that CdLS arises from a NIPBL role independently of its cohesin-loading function. A defect in the recently described loop extrusion function of the cohesin-NIPBL holoenzyme[66–68] could explain most of the defects observed in CdLS (Fig. 7e). Reduced NIPBL function in CdLS-derived cells drives accumulation of cohesin at NIPBL-occupied sites where cohesin is likely loaded, and those cohesin complexes do not move away to reach their final destination at cohesin-CTCF sites. In addition, fewer cohesin peaks colocalize with CTCF, suggesting that an alteration of the NIPBL post-loading function affects cohesin translocation, chromatin loops, or long-range interactions.

Cohesin loop extrusion depends on the presence of NIPBL, not only at the start but throughout the entire process, indicating that cohesin-NIPBL is the active loop extruding holoenzyme[66]. NIPBL's role in chromatin structure is therefore more important than anticipated, and our results suggest that the cohesin-NIPBL function in loop extrusion is affected in CdLS-derived cells. Since NIPBL stimulates cohesin ATPase activity[42,79,80] we propose that mutated NIPBL in CdLS affects cohesin translocation, perhaps by impairing stimulation of the cohesin ATPase activity. This is consistent with ATP hydrolysis being required for relocation of cohesin from the cohesin-loading Scc2 sites in budding yeast[81]. It will be interesting to explore how the CdLS-mutated NIPBL regulates cohesin translocation and loop extrusion formation.

The final cohesin-CTCF sites are mostly constant among tissues and cell types, suggesting that they are stable cohesin binding peaks. By contrast, in CdLS-derived cells, we observed a reduction in the SMC1A signal at cohesin-CTCF sites, in accordance with the less stable chromatin-bound RAD21 observed by iFRAP. Recently, it has been shown that CTCF depletion reduced stable chromatin binding of cohesin-STAG1, suggesting that CTCF stabilizes cohesin-STAG1 on chromatin[82]. Similarly, in CdLS-derived cells, we observed that cohesin is also more dynamic when it does not reach CTCF sites. In addition, RAD21 is released from chromatin at lower salt concentrations in CdLS-derived cells (Fig. 2d), in agreement with the less stably chromatin-bound cohesin found in CdLS-derived cells. High salt concentrations also greatly reduced the number of DNA loops extruded by cohesin-NIPBL[66], suggesting that cohesin entraps DNA non-topologically, or that cohesin and NIPBL interactions are salt-sensitive. Either way, this is consistent with a reduced role for cohesin-NIPBL holoenzyme in loop extrusion in CdLS-derived cells. Reduced processivity of loop extrusion or the ability to spread across chromatin will alter enhancer-promoter interactions and thereby deregulate gene expression.

Around 65% of CdLS patients present pathogenic variants in the NIPBL cohesin loader. We mainly used primary dermal fibroblasts derived from CdLS patients with pathogenic variants in the NIPBL gene. Our calibrated NIPBL ChIP-seq showed a global reduction of the NIPBL signal in CdLS-derived fibroblasts. This may reflect the fact that the mutated NIPBL protein has difficulties binding to DNA, while the wild-type NIPBL allele in the heterozygous patients retains its ability to bind DNA. Alternatively, this might be a consequence of a dominant-negative NIPBL protein struggling to perform its function in cohesin loading, translocating cohesin along the chromatin[62] or in loop extrusion[55,65–67].

Contradictory results were previously reported for NIPBL ChIP-seq data. Some authors reported that a fraction of NIPBL peaks colocalize with cohesin at promoters and enhancers[40,83], while others found mostly no overlap with cohesin, since NIPBL is located at active promoters with no cohesin signal[43,45]. Different affinities among the antibodies, cross-linking problems,

and a lack of calibrated ChIP-seq data could explain many of the discrepancies observed. For this reason, we performed calibrated NIPBL ChIP-seq data after a deep characterization of NIPBL antibodies. We found that NIPBL associates mostly at GC-rich regions, including CpG islands. Promoters, enhancers, and clusters of developmental genes are commonly CpG island-rich regions[76]. Our results reconcile those of previous studies showing that NIPBL colocalizes with promoter and/or enhancer and extend the NIPBL genome-wide distribution to high GC-rich regions.

NIPBL binds at regions with high CG content (>60%), including (but not exclusively) CpG islands around TSS and nearby enhancers. DNA-binding transcription factors and developmentally important genes (including the HOX and PCDH genes) are greatly overrepresented near clusters of three or more CpG islands[76]. We observed that NIPBL colocalized with CpG islands in the developmental gene cluster families rich in CpG islands (Fig. 4g). A large accumulation of NIPBL peaks appear at HOX, protocadherin, histone HIST1, keratin II, IRXA, and IRXB clusters in control cells. Interestingly, reduced NIPBL chromatin association in CdLS-derived cells at HOX and protocadherin clusters correspond with the reduction in the binding of SMC1A in the cohesin-CTCF peaks located nearby and with gene expression deregulation of the HOX and protocadherin gene families. Consistently, during limb development, NIPBL is required for the regulation of long-range interactions and collinear expression of hox genes in zebrafish[84]. Appropriate spatial and temporal expression of Hox genes is important for limb development in all vertebrates[85].

Previous studies reported on the CdLS transcriptome in lymphoblastoid cell lines (LCLs)[86,87]. Only 126 differentially expressed genes were shared by the gene expression profiles of NIPBL-mutated[86] (1431 DEGs) and SMC1A-mutated[87] (1186 DEGs) LCLs, respectively[83]. Similarly, no correlation with our gene expression profile was found.

The transcriptome changes that we observed in CdLS patient-derived fibroblasts were also found in two patients with SMC1A mutations (Supplementary Fig. 4), suggesting that our gene expression profile can be extended to CdLS patients with mutations in other genes. This is consistent with our hypothesis that the cohesin-NIPBL holoenzyme function in loop extrusion is the one most affected in CdLS-derived cells. During the early stages of embryonic development, the defective cohesin-NIPBL in CdLS-mutated cells might provoke an alteration in the chromatin structure, especially in some genomic regions such as those of the developmental gene clusters. These changes in chromatin structures would be fixed in different cell types during development and, for this reason, could be detected in the dermal fibroblast-derived cells.

## Methods

**Cell culture.** Primary dermal fibroblasts from individuals with CdLS and primary dermal fibroblasts from control individuals were kindly provided by Dr J. Pie, Dr A. Latorre-Pellicer, Dr B. Puisac, and Dr. F. Ramos of the Hospital Clínico Universitario "Lozano Blesa" from Zaragoza. The collection sample protocol was supervised and approved by the Clinical Research Ethics Committees of the Hospital Clínico Universitario "Lozano Blesa" from Zaragoza and of the Hospital Universitario de Bellvitge (IDIBELL). The study design and conduct complied with all relevant regulations regarding the use of human study participants and was conducted in accordance with the criteria set by the Declaration of Helsinki. All patients were informed beforehand and gave their written consent to participate in this study. The patients used in this study present pathogenic variants in NIPBL or SMC1A genes (Supplementary Table 1). All cells were routinely tested for mycoplasma by PCR and maintained mycoplasma-free. Cells were grown in Dulbecco's modified Eagle's medium (DMEM + GlutaMAX) supplemented with 10% FBS. Cycling cells were collected 24 h after replating. To obtain quiescent cells, confluent cultures were incubated in DMEM with 0.5% FBS for 24 h. A list of oligonucleotides used in this study is reported in Supplementary Table 3.

**Protein extracts and chromatin fractionation**. For whole-cell extracts, cells were collected by scraping in PBS1x buffer, counted, resuspended in lysis buffer (1% SDS, 10 mM EDTA, 50 mM Tris-HCl pH 8.1), sonicated, and incubated for 1 h on ice. Equal protein concentrations were separated by SDS-PAGE (Mini-Protean TGX Gels 4–12% (Bio-Rad)) and analyzed by immunoblotting.

Protein was fractionated with the Nuclear Extract kit (Active Motif; 40010), following the manufacturer's instructions. To assess the strength of chromatin association of the cohesin complex (Fig. 2d), chromatin fractions were obtained with a Nuclear Extract kit (Active Motif; 40010) treated with buffer A (10 mM HEPES, 1.5 mM $MgCl_2$, 0.34 M sucrose, 10% glycerol, 1 mM DTT, and protease inhibitors) containing 0.25 M or 0.5 M of NaCl for 30 min on ice. Solubilized proteins were separated from insoluble chromatin by low-speed centrifugation (4 min at $1700 \times g$) and washed twice with Buffer A without salt. The insoluble fractions were analyzed by immunoblotting.

**Immunoprecipitation**. Immunoprecipitation of NIPBL was performed as described in ref. [88], briefly $8 \times 10^6$ fibroblasts were resuspended in 0.5 ml lysis buffer (20 mM Hepes pH 7.6, 150 mM NaCl, 10% glycerol, 0.2% NP-0.4, 1 mM NaF, 1 mM sodium butyrate, 1 mM EDTA and complete protease and phosphatase inhibitors (Roche)) and cells were incubated 45 min at 4 °C. Chromatin and soluble fractions were separated by centrifugation at $1000 \times g$ for 3 min at 4 °C. The chromatin pellet was washed 10 times by re-suspension in 1 ml lysis buffer and centrifugation at $1000 \times g$ for 3 min at 4 °C. The chromatin pellet was then resuspended in 250 μl nuclease buffer (lysis buffer complemented with a final concentration of 0.04 units/μl micrococcal nuclease, 0.1 mg/ml RNase A, 20 mM $CaCl_2$, and 0.04 μl/ml Turbo DNase), incubated for 2 h at 4 °C and for 15 min at 37 °C. Chromatin extracts were pre-cleared with magnetic protein-G Dynabeads® (Life Technologies) for 1 h at 4 °C and then the supernatant was incubated with 10 μl of the specific NIPBL antibody (3B9, Novus Biologicals H00025836-M01), and the immunocomplexes were adsorbed onto magnetic protein-G at 4 °C overnight. The beads were washed seven times with buffer B (25 mM Tris-HCl, 500 mM NaCl, 5 mM $MgCl_2$, 0.5% NP-0.4) and eluted with 2X SDS-PAGE loading buffer for 5 min at 95 °C.

**Immunoblotting**. Primary antibodies used in this study were anti-RAD21 1:500 (Abcam, ab992), anti-SMC1A 1:500 (Abcam, ab9262), anti-SMC3 1:500 (Abcam, ab9263), anti-STAG1 1:500 (Bethyl, A302-579A), anti-STAG2 1:200 (Santa Cruz, sc-81852), anti-histone H3 1:1000 (Abcam, ab1791), anti-MEK2 1:500 (BD, 610235), anti-nucleolin 1:500 (Santa Cruz, sc-8031), anti-AcSMC3 1:500 (MBL, PD040), and anti-NIPBL 1:500 (KT55, Abcam, ab106768) for immunoblotting and 5 μl anti-NIPBL (3B9, Novus Biological H00025836-M01), 25 μl H-300 (Santa Cruz, sc-98601), and 25 μl C-9 (Santa Cruz, sc-374625) for immunoprecipitation (Supplementary Figs. 5). Custom-made rabbit polyclonal antibodies against PDS5A (1:500) have been described[89] and those for human sororin (1:500) were raised using the recombinant protein purified from bacteria as antigen. Horseradish peroxidase HRP-conjugated (1:10,000) secondary antibodies (Amersham Biosciences) were used. Immunoblots were quantified using ImageJ software (Fiji v1.53).

**Immunofluorescence**. Human fibroblasts were cultured on coverslips and fixed with paraformaldehyde 4% for 15 min at room temperature, washed three times with 1× PBS, and permeabilized with 0.1% of Triton X-100 in PBS for 20 min at RT. The coverslips were washed again and blocked with 1% BSA in PBS for 1 h at RT. After blocking, cells were incubated overnight at 4 °C with the corresponding antibody diluted in blocking solution (anti-NIPBL 3B9) 1:100, anti-SMC1A[73] 1:1000, and anti-RAD21 1:100. Cells were incubated with the secondary antibody, Cy3-labeled α-mouse (GE Healthcare) for anti-NIPBL and Alexa 555-labeled α-rabbit (Abcam, ab150082) for anti-SMC1A and anti-RAD21, 1:1000 for 1 h at RT. The coverslips were air-dried and 3 μl of the mounting medium with DAPI (Vectashield, H-1200) were added. Images were captured with a Zeiss Axio Observer Z1 inverted epifluorescence microscope with an Apotome system equipped with an HXP 120C fluorescent lamp and a Carl Zeiss Plan-Apochromat 63× N.A 1.40 oil objective, in conjunction with ZEN software (blue edition, v1.0). Fluorescence was quantified using ImageJ software (Fiji v1.53).

**Photobleaching experiments: inverse fluorescence recovery after photobleaching (iFRAP)**. Human fibroblasts were nucleofected 18 h before acquiring microscope images with the plasmid EGFP-RAD21[71] using the Amaxa® Human Dermal Fibroblasts Nucleofector Kit (Lonza), following the manufacturer's instructions. Cells were grown on glass plates. One hour before imaging, the medium was changed to one without phenol red, and 100 μg/ml of cycloheximide was added to reduce synthesis of new GFP-tagged cohesin for experiments lasting longer than 1 h. All iFRAP experiments used ten interactions of photobleaching at 100% transmission of 488 nm laser of the Carl Zeiss LSM880 confocal microscope, with a 63× NA 1.4 objective with ZEN software (black edition v2.3). Cytoplasmic and nuclear regions were bleached, leaving only half of the nuclear region unbleached. The first post-bleach frame was acquired 60 s after photobleaching to allow for complete equilibration of unbleached soluble GFP-cohesin across the nucleus. Fluorescence recovery after bleaching was quantified using ImageJ

software (Fiji v1.53), considering the difference in mean intensity between bleached and unbleached regions over 1 h. The drop in mean fluorescence intensity in the unbleached nuclear region after photobleaching was used to calculate the chromatin-bound fraction of EGFP-RAD21[71].

**Gene expression microarrays and analysis**. Total RNA from the early passage (p3) of dermal-derived fibroblasts from four healthy individuals were used as controls and five different NIPBL-mutated CdLS patients, including two patients in duplicate, were analyzed using Agilent SurePrint G3 Human Gene Expression Microarrays v2 (ID 039494). The two-color labeling protocol for Microarray-Based Gene Expression Analysis v. 6.5 (Agilent) was used. Agilent Feature Extraction Software (FES 10.7.3.1) was used to read and process the microarray image files. Raw data are summarized in Supplementary Data 1.

Raw data import and pre-processing steps were done with the limma package (v4.44.3)[90] in R (v4.0.2). The raw signal data were background corrected with the $normexp + offset$ method considering a $k$ value of 50 and normalized between arrays using the quantile-normalization method. The selection of differential expressed genes (DEG) was based on a linear model adjusted by age stages. Two age stages were considered, individuals under and over 15 years old. $p$-values were computed using moderated $t$-statistics using empirical Bayes shrinkage and adjusted to control the false discovery rate using the Benjamini and Hochberg method. Genes were considered DEG when FDR < 0.25 and |logFC|>0.5.

*Gene Ontology analysis*. We used DAVID bioinformatics[91] and pre-ranked GSEA (v4.1.0)[92] using the biological processes curated in the collection C5 from MsigDB[93] to search for enriched gene ontology biological processes. The log-fold change values resulting from the DE analysis were used to rank the list of genes. The GSEA performance was run using 1000 gene permutations, a weighted gene enrichment statistic, and gene sets were limited to those containing between 15 and 500 genes. Gene sets with an FDR $q$-value lower than 0.25 were considered significantly enriched.

**RNA purification and quantitative real-time PCR (qRT-PCR) analysis**. cDNAs were prepared with the Transcriptor First Strand cDNA Synthesis Kit (Roche) from total RNA from passages p4-p8 (NucleoSpin RNA Kit, Macherey-Nagel), and qRT-PCR analyses were performed using the SYBR Premix Ex Taq (Takara) and a Light Cycler 480 instrument (Roche). AnyGenes custom panels (BioNova) were used to validate some gene expression results from the microarrays. Expression was normalized with respect to the mean level of expression of the housekeeping genes ACTB1, TFAP2E, ACBD3, and LETMD1. Reactions were performed in triplicate.

To investigate the effect in gene expression upon introduction of control NIPBL (Supplementary Fig. 3), patient 3-derived fibroblast were nucleofected (Amaxa® Human Dermal Fibroblasts Nucleofector Kit (Lonza)) with 2 μg of the commercial plasmid IDN3 (NIPBL-GFP) (NM_015384) Human Tagged ORF Clone (Origene) in triplicate, followed by qRT-PCR after 24 h.

**Chromatin immunoprecipitation followed by sequencing and analysis**. For SMC1A ChIP-seq, two independent ChIP experiments were performed in two NIPBL-mutated CdLS individuals (Patient 3 and Patient 4) and one healthy individual using anti-SMC1A rabbit polyclonal antibody[73]. Cells were cross-linked with 1% formaldehyde, which was added to the medium for 15 min at room temperature. After a quenching step with 0.125 M glycine, fixed cells were washed twice with 1× PBS, pelleted, and lysed in lysis buffer (1% SDS, 10 mM EDTA, and 50 mM Tris-HCl, pH 8.1) containing 1 μM PMSF and protease inhibitors (Roche) for 1 h at 4 °C. Cells were then sonicated using the Covaris system (shearing time 20 min, 20% duty cycle, intensity 10, 200 cycles per burst, and 30 s per cycle) in a 1 ml volume. Magna-beads A + G (Pierce) and 8 μl of antibody against SMC1A were bound for 3–4 h at 4 °C in PBS, added to the chromatin (250 μg diluted four times with buffer TE 0.5% SDS) and incubated overnight at 4 °C. Immunoprecipitated chromatin was washed three times with wash buffer1 (50 mM Hepes pH 7.5, 500 mM NaCl, 1 mM EDTA, 1% Triton X-100, 0.1% sodium deoxycholate, 0.1% SDS, 1 mM PMSF) and twice with wash buffer2 (10 mM Tris pH 8, 0.25 M LiCl, 0.5% NP-0.4, 0.5% sodium deoxycholate, 1 mM EDTA, 1 mM PMSF). The DNA was eluted from beads with 200 μl of elution buffer (1% SDS, 10 mM EDTA and 50 mM Tris-HCl, pH 8.1) and incubated for 45 min at 65 °C. To carry out reverse-cross-linking the supernatant was incubated for 4 h at 65 °C, and then treated with 7.5 μl of proteinase K (20 mg/ml) and 1 μl of RNAse A (Sigma) for 1 h at 37 °C. DNA was purified using the NucleoSpin Gel and PCR Clean-up kit (Macherey-Nagel). Around 5 ng of immunoprecipitated chromatin in each sample was used for library preparation.

For NIPBL ChIP-seq, cycling cells from Patient 3 CdLS-derived fibroblasts and one healthy control were used for calibrated ChIP-seq (Spike-in ChIP-seq) experiments. For the spike, 5% of sonicated chromatin from mouse embryonic stem (ES) cells was added to the human chromatin. Protein-G beads (70 μl, Life Technologies) and 10 μl of antibody against NIPBL (3B9, Novus Biological H00025836-M01) were used for immunoprecipitation.

ChIP-seq fastq files were trimmed with Trim Galore (v0.4.5) and aligned to the hg19 reference genome with Bwa mem (v0.7.17) and Samtools (v1.7). Reads with

mapping quality ≤3 were removed. Duplicated reads were also removed using the Picard (v2.8.15) tool MarkDuplicates. MaNorm (v1.0) software was used to identify differential enrichment regions between Patient and Control samples. A significance threshold of −log10 adjusted value of $p > 2$ was used to identify regions differentially enriched between samples.

The mouse spike-in chromatin was used to normalize across NIPBL samples[55]. Briefly, reads were first aligned to a combined reference genome consisting of the human hg19 and the mouse mm10 genome assemblies. Then, for each sample, those reads mapping uniquely to mm10 were quantified. Since the proportion of mouse spike-in DNA was the same across samples, the differences in mm10 read coverage between samples was used to normalize the NIPBL sample signal. Specifically, the bam files with the greatest number of uniquely mapped reads to mm10 were downsampled so that all samples would yield the same amount of unique mm10 reads. The R (v3.4.4) package NCIS was used to calculate a scaling factor between IP and INPUT samples that was used by the peak calling software Macs2 (v2.1.1.20160309) to sensitively call peaks across all bam files. A threshold of LFC > 5 was used to filter out dubious NIPBL peaks.

All peaks were annotated using Homer (v3.12) software and custom R scripts. Specifically, the distance to TSS and gene names were annotated using Homer, and the distance to CpG and Enhancers were annotated using R's GRanges methods. CpG island information was obtained from UCSC queries using rtracklayer and Enhancer information was obtained from the Enhancer Atlas database (http://www.enhanceratlas.org/).

ChIP signal at specific genomic locations was obtained using custom scripts in R. Specifically, for each called peak, a ±500 bp window was opened around the peak's midpoint. The generated data were visualized in R using custom scripts. Venn diagrams were generated using the online resource http://bioinformatics.psb.ugent.be/webtools/Venn/.

**ChIP-seq validation by ChIP-qPCR**. Cells were collected, cross-linked, and sonicated as described above and chromatin immunoprecipitation was performed following the protocol of the iDeal ChIP-seq kit for transcription factors (Diagenode). The immunoprecipitated DNA was used to perform qPCR with the SYBR Premix Ex Taq (Takara) and the Light Cycler 480 instrument (Roche). The relative amount of each amplified fragment was normalized with respect to the amplification obtained from input DNA. Primers were designed using the Primer3 web site[94].

**DNA methylation analysis**. Genomic DNA was obtained from CdLS-derived fibroblasts with a NucleoSpin DNA RapidLyse Kit (Macherey-Nagel). All DNA samples were quantified by the fluorometric method (Quant-iT PicoGreen dsDNA Assay, Life Technologies), and assessed for purity by NanoDrop (Thermo Scientific) 260/280 and 260/230 ratio measurements. Six hundred nanograms of DNA were processed using the EZ-96 DNA Methylation kit (Zymo Research) following the manufacturer's recommendations for Infinium assays.

Seven microliters of DNA were processed following the Illumina Infinium HD DNA Methylation Assay Protocol[95]. Infinium 450 K DNA Methylation Array was hybridized. Data were analyzed with R (version 3.4.2) using minfi (1.23.4). Quantile and functional normalization were performed using the minfi[96] package, following the author's guidelines. The DNA methylation score of each CpG is represented as a β-value. We excluded possible sources of biological and technical biases that could have affected the results (probes located on X/Y chromosomes, SNPs, etc.). We also evaluated the detection probabilities (comparing signals intensities against background noise) for all CpGs and excluded those CpGs with values of $p > 0.01$ in more than one sample. For the differential DNA methylation analysis, a linear model was derived using minfi in R for all the CpGs. The resulting probabilities were corrected for multiple testing (FDR).

DNA Methylation was validated by bisulfite pyrosequencing (PyroMark Q96 System). Five DMPs for HOXC4, 5, 6 and seven DMPs for HOXB3 were tested. Means and SEMs were calculated for all the DMPs within a promoter and represented as DMRs.

**In silico intra-TAD loops prediction**. We adapted the computational method proposed by Matthews and Waxman[77] to predict TADs from ChIP-seq data. This tool focuses on two main TADs characteristics: (i) TAD boundaries or anchors are enriched with both CTCF and cohesin binding and (ii) the convergent orientation of both TAD CTCF anchors. To identify CTCF motifs in ChIP-seq regions and their corresponding orientation we used FIMO program (v5.1.0) using ChIP-seq GEO data for CTCF (GSM935404). Our experimental control and two patients (Patient 3 and Patient 4) SMC1A ChIP-seq datasets were used. Computational tool was executed with the minimum proportion default value (40%). This parameter defines the minimum cohesin and CTCF anchors to be considered for identifying a first list of putative loops. Resulting TADs lists were reduced after removing redundant TADs. Two TADs were considered redundant if they shared the same TAD upstream anchor and showed a minimum reciprocal overlapping of 98%. The wider TAD was kept in the final TADs list. Prediction tool scored TADs based on the respective Macs2 ChIP-seq peak strength and FIMO CTCF motif occurrence score in regions of interest. To assign combined TAD scores, we combined cohesin and CTCF ChIP-seq scores by means of their geometric mean.

To compare the in silico prediction tool with experimental published TADs we used the high-resolution Hi-C data reported by Rao et al.[78] in IMR90 cell lines (GEO dataset GSE63525). We scanned all the loop anchors searching for a CTCF motif (JASPAR[4] motif MA0139.1) using FIMO as in the prediction method selecting the experimental loops with CTCF motif at both anchors. We quantified the in silico and experimental TADs that reciprocally overlapped in a 95% minimum and categorized them as a complete match. We considered a partial match when (a) an in silico TAD fully included an experimental TAD (include) or vice versa (inside), and (b) in silico and experimental TADs partially overlapped. The rest of in silico or experimental TADs were labeled as unmatched TADs. Given a TAD, we selected the partial match with the highest overlap percentage if several hits were retrieved. Next, we compared the in silico prediction tool in control and CdLS-derived samples. We categorized TADs derived from the control sample as complete match, partial match, or unmatched TAD as above and compared them to the integrated patients TADs list. TADs scores distributions were compared among the three categories and statistically tested using a Wilcoxon test.

For comparison and visualization purposes, we used bedtools (v2.26.0) utilities, phyton (v2.7.17), ChIPpeakAnno (v3.18.1), GenomicRanges (v1.36.0), ggplot2 (v3.3.2), ggubr (v0.2.3), and Bioconductor (v3.9.0) R (v3.6.3) packages.

**3C-qPCR studies**. The chromosome conformation capture assay was performed as described in Williamson et al.[97] with some modifications. Approximately $8 \times 10^6$ cells were fixed with 1% formaldehyde for 10 min at room temperature. Cross-linking was stopped with 125 mM glycine for 5 min followed by two washes with cold PBS. Cells were centrifuged at $1000 \times g$ for 5 min at 4 °C, supernatants were removed, and cell pellets were flash-frozen on dry ice. Cells were incubated for 30 min at 4 °C in 200 µl of lysis buffer (10 mM Tris pH 8.0, 10 mM NaCl, 0.2% NP-40, supplemented with Complete protease inhibitor cocktail (Roche)). Chromatin was then centrifuged at $2000 \times g$ for 5 min. Supernatants were removed, pellets were washed twice with 100 µl of 1X HindIII buffer (New England Biolabs), and the chromatin pellet was resuspended in 400 µl of 1X HindIII buffer and incubated for 10 min at 65 °C with 0.1% SDS. Forty-four microliters of 10% Triton X-100 was added before overnight digestion with 400 U of HindIII at 37 °C. The restriction enzyme was then inactivated by adding 86 µl of 10% SDS and incubating for 30 min at 65 °C. Samples were then diluted into 7.5 ml of ligation mix (750 µl of 10% Triton X-100, 750 µl of 10X ligation buffer, 80 µl of 10 mg/ml of BSA, 80 µl of 100 mM ATP, 3000 cohesive end units of T4 DNA ligase) and incubated for 2 h at 16 °C and 30 min at RT. 3C libraries were incubated overnight at 65 °C with 25 µl of Proteinase K (20 mg/ml) and an additional 25 µl of Proteinase K the following day for 2 h. The DNA was purified by two phenol–chloroform extractions and precipitated with 0.1 vol of 3 M NaOAc (pH 5.2) and 2.5 vol of cold EtOH. After at least 1 h at −80 °C, the DNA was centrifuged at $20,000 \times g$ for 25 min at 4 °C, and the pellets were washed with cold 70% EtOH twice. DNA was resuspended in 400 µl of TE (pH 8.0) and transferred to 1.5 ml tubes for another phenol–chloroform extraction and precipitation with 40 µl of 3 M NaOAc (pH 5.2) and 1.1 ml of cold EtOH. DNA was recovered by centrifugation and washed five times with cold 70% EtOH. Pellets were then dissolved in 100 µl of TE (pH 8.0) and incubated with 1 µl of 10 mg/ml RNase A for 15 min at 37 °C. DNA was used to perform a qPCR reaction to validate six predicted TADs: TAD1 (chr6:39833046–3986 0745), TAD2 (chr7:30588369–30780468), TAD3 (chr3:48506691–48647407), TAD4 (chr19:41055101–41140816), TAD5 (chr4:8268881–8405291), and TAD6 (chr12:54 399697–54601972). Four 100 bp genomic regions without HindIII restrictions sites inside were used to normalize the data.

**Statistical analysis**. Statistical significance was tested by analysis of the unpaired two-tailed Student's t-test (GraphPad Prism (v5)). Values of $p < 0.05$ were considered statistically significant, ****$p < 0.0001$, ***$p < 0.001$, **$p < 0.01$, *$p < 0.05$.

## Data availability
The data that support this study are available from the corresponding authors upon reasonable request. ChIP-seq data generated in the course of this work have been deposited in the Gene Expression Omnibus (GEO) under the accession number GSE145966. Encode data for RAD21 (ENCFF361FUX) and MNase (ENCFF000VLK), GEO data for CTCF (GSM733672) and DNase (GSM816655), and Enhancer Atlas database for enhancer information (http://www.enhanceratlas.org/) were used for Fig. 5 and Supplementary Fig. 7. GEO data for CTCF (GSM935404) and the high-resolution Hi-C data for IMR90 cell lines (GSE63525) were used for the intra-TAD prediction tool for Fig. 7 and Supplementary Fig. 8. Other data supporting the findings of this study are available within the paper, its supplementary information files, and the source data file. Source data are provided with this paper.

## Code availability
The custom scripts generated during the current study are available from https://bitbucket.org/qgenomics/smc1a-nipbl.

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

# ARTICLE

63. Collier, J. E. et al. Transport of DNA within cohesin involves clamping on top of engaged heads by SCC2 and entrapment within the ring by SCC3. *Elife* **9**, e59560 (2020).

64. Rhodes, J., Mazza, D., Nasmyth, K. & Uphoff, S. Scc2/Nipbl hops between chromosomal cohesin rings after loading. *Elife* **6**, e30000 (2017).

65. Haarhuis, J. H. I. et al. The cohesin release factor WAPL restricts chromatin loop extension. *Cell* **169**, 693–707.e14 (2017).

66. Davidson, I. F. et al. DNA loop extrusion by human cohesin. *Science* **366**, 1338–1345 (2019).

67. Kim, Y., Shi, Z., Zhang, H., Finkelstein, I. J. & Yu, H. Human cohesin compacts DNA by loop extrusion. *Science* **366**, 1345–1349 (2019).

68. Golfier, S., Quail, T., Kimura, H. & Brugués, J. Cohesin and condensin extrude DNA loops in a cell-cycle dependent manner. *Elife* **9**, e53885 (2020).

69. Chao, W. C. H. et al. Structure of the cohesin loader Scc2. *Nat. Commun.* **8**, 13952 (2017).

70. Kikuchi, S., Borek, D. M., Otwinowski, Z., Tomchick, D. R. & Yu, H. Crystal structure of the cohesin loader Scc2 and insight into cohesinopathy. *Proc. Natl Acad. Sci. USA* **113**, 12444–12449 (2016).

71. Gerlich, D., Koch, B., Dupeux, F., Peters, J. M. & Ellenberg, J. Live-cell imaging reveals a stable cohesin-chromatin interaction after but not before DNA replication. *Curr. Biol.* **16**, 1571–1578 (2006).

72. Li, Y. et al. The structural basis for cohesin–CTCF-anchored loops. *Nature* **578**, 472–476 (2020).

73. Remeseiro, S., Cuadrado, A., Gómez-Lǎpez, G., Pisano, D. G. & Losada, A. A unique role of cohesin-SA1 in gene regulation and development. *EMBO J.* **31**, 2090–2102 (2012).

74. Kawauchi, S. et al. Multiple organ system defects and transcriptional dysregulation in the Nipbl+/- mouse, a model of Cornelia de Lange syndrome. *PLoS Genet.* **5**, e1000650 (2009).

75. Del Campo, M. et al. Monodactylous limbs and abnormal genitalia are associated with hemizygosity for the human 2q31 region that includes the HOXD cluster. *Am. J. Hum. Genet.* **65**, 104–110 (1999).

76. Branciamore, S., Chen, Z. X., Riggs, A. D. & Rodin, S. N. CpG island clusters and pro-epigenetic selection for CpGs in protein-coding exons of HOX and other transcription factors. *Proc. Natl Acad. Sci. USA* **107**, 15485–15490 (2010).

77. Matthews, B. J. & Waxman, D. J. Computational prediction of CTCF/cohesin-based intra-TAD loops that insulate chromatin contacts and gene expression in mouse liver. *Elife* **7**, e34077 (2018).

78. Rao, S. S. P. et al. A 3D map of the human genome at kilobase resolution reveals principles of chromatin looping. *Cell* **159**, 1665–1680 (2014).

79. Murayama, Y., Samora, C. P., Kurokawa, Y., Iwasaki, H. & Uhlmann, F. Establishment of DNA-DNA interactions by the cohesin ring. *Cell* **172**, 465–477 (2018).

80. Shi, Z., Gao, H., Bai, X. C. & Yu, H. Cryo-EM structure of the human cohesin-NIPBL-DNA complex. *Science* **368**, 1454–1459 (2020).

81. Hu, B. et al. ATP hydrolysis is required for relocating cohesin from sites occupied by its Scc2/4 loading complex. *Curr. Biol.* **21**, 12–24 (2011).

82. Wutz, G. et al. ESCO1 and CTCF enable formation of long chromatin loops by protecting cohesinSTAG1 from WAPL. *Elife* **9**, e52091 (2020).

83. Boudaoud, I. et al. Connected gene communities underlie transcriptional changes in Cornelia de Lange syndrome. *Genetics* **207**, 139–151 (2017).

84. Muto, A. et al. Nipbl and mediator cooperatively regulate gene expression to control limb development. *PLoS Genet.* **10**, e1004671 (2014).

85. Zakany, J. & Duboule, D. The role of Hox genes during vertebrate limb development. *Curr. Opin. Genet. Dev.* **17**, 359–366 (2007).

86. Liu, J. et al. Transcriptional dysregulation in *NIPBL* and cohesin mutant human cells. *PLoS Biol.* **7**, e1000119 (2009).

87. Mannini, L. et al. Mutant cohesin affects RNA polymerase II regulation in Cornelia de Lange syndrome. *Sci. Rep.* **5**, 16803 (2015).

88. Holzmann, J. et al. Absolute quantification of cohesin, CTCF and their regulators in human cells. *Elife* **8**, e46269 (2019).

89. Carretero, M., Ruiz-Torres, M., Rodríguez-Corsino, M., Barthelemy, I. & Losada, A. Pds5B is required for cohesion establishment and Aurora B accumulation at centromeres. *EMBO J.* **32**, 2938–2949 (2013).

90. Ritchie, M. E. et al. Limma powers differential expression analyses for RNA-sequencing and microarray studies. *Nucleic Acids Res.* **43**, e47 (2015).

91. Huang, D. W., Lempicki, R. A. & Sherman, B. T. Systematic and integrative analysis of large gene lists using DAVID bioinformatics resources. *Nat. Protoc.* **4**, 44–57 (2009).

92. Subramanian, A. et al. Gene set enrichment analysis: a knowledge-based approach for interpreting genome-wide expression profiles. *Proc. Natl Acad. Sci. USA* **102**, 15545–15550 (2005).

93. Liberzon, A. et al. Molecular signatures database (MSigDB) 3.0. *Bioinformatics* **27**, 1739–1740 (2011).

94. Untergasser, A. et al. Primer3-new capabilities and interfaces. *Nucleic Acids Res.* **40**, e115 (2012).

95. Sandoval, J. et al. Validation of a DNA methylation microarray for 450,000 CpG sites in the human genome. *Epigenetics* **6**, 692–702 (2011).

96. Aryee, M. J. et al. Minfi: a flexible and comprehensive Bioconductor package for the analysis of Infinium DNA methylation microarrays. *Bioinformatics* **30**, 1363–1369 (2014).

97. Williamson, I. et al. Spatial genome organization: contrasting views from chromosome conformation capture and fluorescence in situ hybridization. *Genes Dev.* **28**, 2778–2791 (2014).

## Acknowledgements

We thank all the members of our laboratory for discussing this work and for their critical reading of the manuscript. We thank Diana Garcia for her technical support. We thank the CERCA Program/Generalitat de Catalunya for institutional support. This work was funded by the Spanish Ministry of Economy, Industry and Competitiveness (MINECO), which is part of the State Agency, through the projects BFU2013-43132-P and BFU2016-77975-R to E.Q., and BFU2016-79841 to A.L. (co-funded by the European Regional Development Fund, ERDF, a way to build Europe) and the MaratóTV3 (101030 FMTV3) to E.Q. A.L.-P., B.P., F.R., and J.P. are funded by the Spanish Ministry of Health ISCIII-FIS (Ref. PI19/01860) and by the Gobierno de Aragon (Ref. B32_17R) and belong to the GCV02 of CIBERER. J.L.M. is a recipient of a grant from the Instituto Carlos III, which is part of the State Agency, grant number CA18/00045 (co-funded by the European Regional Development Fund, ERDF, a way to build Europe) and M.M. is a recipient of the fellowship "Personal Técnico de Apoyo" number PTA2018-016371-I funded by the Spanish Ministry of Science, Innovation and Universities (MCIU), which is part of the State Agency.

## Author contributions

P.G., R.F.-H., and E.Q. designed and performed the experiments. A.C. and A.L. helped with the SMC1A ChIP-sequencing experiments, antibodies, and reagents. J.L.M. analyzed the gene expression data. J.R. and I.C. performed the ChIP-seq bioinformatics analysis and helped to interpret the data. J.S., A.G., and M.E. carried out the DNA methylation studies. M.M. and J.L.M. carried out the intra-TAD loops prediction studies. A.L.-P., B.P., F.R., and J.P. isolated the human dermal fibroblasts from patients and controls, verified the pathogenic variants, and provided the genotype-phenotype information of the CdLS patients. P.G., R.F.-H., and E.Q. interpreted the data. E.Q. wrote the manuscript. All authors reviewed and discussed the manuscript.

## Competing interests

The authors declare no competing interests.
