## [Peer Review File · Nature Communications]

REVIEWER COMMENTS

Reviewer #1 (Remarks to the Author):

Garcia, Fernandez-Hernandez and Queralt's group has investigated the genes that show altered expression in CdLS patient-derived cells with mutations in Nipbl a major gene found to be mutated in the majority of CdLS. Furthermore, they studied the altered occupancy of the Nipbl and Smc1A protein genome-wide. Major findings show the altered expression level of HOX genes, protocadherin and other developmental genes, which are known to be important for regulation of normal development. Results provide explanation for the etiology of CdLS. Patient cells also shows reduced Nipbl and Smc1A binding at HOX and PCDH gene clusters. Overall it is an important contribution to the CdLS field and also provides insights into how mutations in cohesin and protein associated with cohesin lead to disorders with skeletal and limb abnormalities. CdLS is a neurodevelopmental disorder, although fibroblasts are useful models to study CdLS questions remain how do these mutations affect the development of the nervous system as it is the major system affected in CdLS?

Specific comments that need to be addressed in the revised version.

- How are the skin fibroblasts derived from patients and controls? Which part of the body they are derived from? A different set of HOX genes are known to be expressed at different body axis, it is important to make sure the skin fibroblast cells comes from the same body part. Also how many control cell lines are used are they originated form different people?
- ChIPseq and RNAseq analysis do not convincingly show the genes that are misregulated directly due to NIPBL mutations or due to indirect consequence, for e.g. altered genes could be due to altered proliferation of particular cell type over the other cell types.
- To strengthen the conclusion authors could perform a depletion experiment for NIPBL or create a patient mutation in control cell lines and compare if they can get similar genes that show altered expression. Can they rescue altered gene expression with wild type NIPBL?
- Although Nipbl levels are down in the HOXD cluster which is consistent with the reduced expression of HOXD genes.
- HOXB and C are hugely upregulated in patients but this is not discussed in the paper. Are they affected due to the altered expression of HOXD?
- Nipbl ChIPseq data was shown in patient 3, what about other patients? Are the peaks that lost in patient 3 are consistent with other patient cells? The use of independent patient data will strengthen the conclusion drawn about the common genes that are dysregulated.

Reviewer #2 (Remarks to the Author):

This manuscript by Garcia et al. investigates transcriptional and cohesin localisation changes in primary fibroblasts from patients suffering from Cornelia de Lange syndrome (CdLS), a developmental disorder that can affect many parts of the body and is associated with mutations of the cohesin loader factor NIPBL. The authors report that cohesin levels are not affected in CdLS patient-derived fibroblasts, however cohesin stability on chromosomes is reduced. Transcriptional analysis of these cells revealed alterations in developmental genes suggesting that the regulation of these genes might not be tightly regulated during development/differentiation in these individuals.

The authors then studied the distribution of NIPBL and cohesin on the genome of the CdLS fibroblasts by Chip-seq and reported a reduction in NIPBL sites at promoters. Cohesin was found to be increased at sites

occupied by NIPBL but reduced at CTCF sites suggesting that cohesin translocation is impaired. This is consistent with recent data demonstrating a role for NIPBL stimulating cohesin ATPase activity and loop extrusion.

Overall, this study is well executed and contains a wealth of information that will be of great interest to scientists in the field, however I would suggest that the authors perform HiC analysis on CdLS fibroblasts and support the changes observed in their current ChIP-seq analysis with the expected alterations that this should cause in genome architecture in these cells. Besides the HiC analysis I don't have any major concerns that need to be addressed.

Reviewer #3 (Remarks to the Author):

The manuscript by Garcia et al. presents a study on transcriptome analysis of CdLS patients' derived fibroblasts. It also reports analysis of genome-wide distribution and CpG islands. The topic is clearly interesting, and it goes beyond the chromatinopathies field as cohesins and their regulation are fundamental in cell biology.

Despite the clear importance of such studies, a number of inaccuracies must be addressed before possibly considering the work for publication.

First and foremost, the bulk of the study is based on primary fibroblasts passaged 2 to 6 times. This is very troubling as, especially considering the low number of samples, the same exact passage should have been used for comparison. Moreover, only 2 healthy donors have been used. This is not acceptable. No information on genetics of this healthy donors is reported.

Further:

1-Western Blot analysis must be revised: quantification should be used for saying "similar" or "more". The methods do not explain if results presented have been obtained by one blot with 5-7 antibodies, by stripping 7-8 times or by single blotting with actin. In the first case, which could lead to cross-reaction, the full membrane should be presented. In the second case, quantifying a multiple-stripped membrane could lead to mistake. In the last case, each experiment should be presented with the specific control (at least in supplementary material). Finally, how many times were WB repeated? How many experimental and how many technical controls?

2-For immunofluorescence experiments: similar is not accurate. Again, this needs replicates, quantification and stats.

3-Photobleaching: was it performed on one patient vs on control? Only once? Replicates and stats are missing.

4-The transcriptome analysis is very confusing: if fibroblasts were used, why developmental genes (including skeletal and urogenital) should be downregulated? Such pathways should not be different in the used cell lines. Moreover, how can author speculate in the results and discussion sections about implication of their findings during development and CdLS phenotype using differentiated primary cells? I believe that the modelling system does not support authors' interpretation.

5-A general lack of accuracy is noted: nomenclature of genes and proteins is wrong throughout the text. Please refer to HUGO guidelines. Methods are not detailed. A number of previously published papers on the topic and with similar experiments are missing. Please carefully assess the literature and bibliography. Figures are difficult to follow: please use panels, arrows and a general order in figure preparation. Finally, figure legends should explain everything in the figure, not just the main finding. Please revise figure legends thoroughly.

REVIEWER COMMENTS

Globally, the main changes of the manuscript are:

- New Supplementary Figure 3, rescue of the altered gene expression in the CdLS-derived cells by re-introduction of wild-type *NIPBL*.
- A new gene expression analysis including 4 controls derived-fibroblasts have been included in new Figure 3.
- New Nipbl ChIP-qPCR data using three new patients is included (new supplementary Fig. 6).
- New data using an *in silico* intra-TAD prediction tool (as described in Matthews and Waxman 2018 eLife, 7:e34077) and validation by 3C-qPCR experiments (new Figure 7 and supplementary Figure 8)
- To strengthen our gene expression data, the validation of the HOX and protocadherin gene expression changes has been study in a second SMC1A mutated patient (Supplementary Figure 4).

We really think that our new data strengthen the conclusions of our manuscript and it is in line with the newly published data during the last months suggesting Nipbl regulation of cohesin's ATPase activity, loop extrusion and DNA entrapment and translocation (Golfier et al., eLife 2020; Shi et al., Science 2020; Collier et al., eLife 2020). These recently published papers have been cited in the current version of the manuscript.

Reviewer #1 (Remarks to the Author):

Garcia, Fernandez-Hernandez and Queralt's group has investigated the genes that show altered expression in CdLS patient-derived cells with mutations in Nipbl a major gene found to be mutated in the majority of CdLS. Furthermore, they studied the altered occupancy of the Nipbl and Smc1A protein genome-wide. Major findings show the altered expression level of HOX genes, protocadherin and other developmental genes, which are known to be important for regulation of normal development. Results provide explanation for the etiology of CdLS. Patient cells also shows reduced Nipbl and Smc1A binding at HOX and PCDH gene clusters. Overall it is an important contribution to the CdLS field and also provides insights into how mutations in cohesin and protein associated with cohesin lead to disorders with skeletal and limb abnormalities. CdLS is a neurodevelopmental disorder, although fibroblasts are useful models to study CdLS questions remain how do these mutations affect the development of the nervous system as it is the major system affected in CdLS?

We thank the reviewer for the comments and suggestions. We agreed with the reviewer that further work is needed to study how the CdLS mutations affect the development of the nervous system and it is a critical point we want to study in the future.

Specific comments that need to be addressed in the revised version.

- How are the skin fibroblasts derived from patients and controls? Which part of the body they are derived from? A different set of HOX genes are known to be expressed at different body axis, it is important to make sure the skin fibroblast cells comes from the same body part. Also how many control cell lines are used are they originated form different people?

The primary dermal fibroblasts from individuals with CdLS and primary dermal fibroblasts from control individuals were kindly provided by our clinician collaborators: Dr J. Pie, Dr B. Puisac, Dr. Ana Latorre and Dr F. Ramos of the Hospital Clínico Universitario "Lozano Blesa" from Zaragoza. They are all derived from dermal biopsies from the arms.

As indicated in Table S1, the control cell lines are derived from 2 healthy controls, a boy and a girl of similar ages to the CdLS patients. Originally, the experiment was designed including 4 controls (as indicated in the nr-report submitted with the original manuscript). To obtain control samples from healthy children was quite complicated since they have to be related to the same Hospital and undergo some kind of surgery to be willing to provide the samples and the ethical consent. For this reason, in order to have more controls we included two adult controls (parents of CdLS Patient 4), since relatives of the CdLS patients are easier to recruit. Initial PCA and clustering analyses of the gene expression data showed a clear batch effect due to the age (samples >15 years old clustered together). For this reason, initially we carried out the analysis using only the two controls children derived-fibroblasts in our original manuscript.

Nevertheless, we have now included the gene expression data including the 4 controls samples. We have introduced a batch effect correction for the age after normalization of the data. Similar results are obtained and we also observed deregulation of the *HOX* and protocadherin genes using the 4 controls (new Figure 3). Just note, that we avoid to performed deep comparison on DEG's since we are aware that the final list of DEG's might vary depending on the controls used.

- ChIPseq and RNAseq analysis do not convincingly show the genes that are misregulated directly due to NIPBL mutations or due to indirect consequence, for e.g. altered genes could be due to altered proliferation of particular cell type over the other cell types.

To strengthen the conclusion authors could perform a depletion experiment for NIPBL or create a patient mutation in control cell lines and compare if they can get similar genes that show altered expression. Can they rescue altered gene expression with wild type NIPBL?

We thank the reviewer for these suggestions. We agree with the reviewer that -omics experiments do not distinguish among primary and secondary effects.

Interestingly, we can rescue the altered gene expression of CdLS-derived cells by re-introduction of wild type *NIPBL* in genes related to embryonic and central nervous system development, such as the protocadherin, the *HOXC* and *HOXD* genes (new Supplementary Figure 3). This suggests that at least some of the embryonic and central nervous system deregulated genes observed in CdLS-derived cells are directly due to *NIPBL* mutations.

Next, we wondered whether *NIPBL* depletion experiment provokes similar changes in gene expression. By using siRNA for *NIPBL*, we could not detect any changes in gene expression in the *HOX* and protocadherin genes by RT-qPCR experiments (see figure below). We cannot rule out that the remaining Nipbl protein detected by western blot (Figure Reviewer 1) is still able to bind to chromatin and perform its function. Moreover, this result suggests that haploinsufficiency of *NIPBL* is not enough to generate the altered gene expression observed in CdLS patients. However, since this is a negative result and more experiments will be needed to argue that the CdLS patients are more than a mere haploinsufficiency of *NIPBL* we did not include this experiment in the manuscript.

Figure Reviewer 1. Partial depletion of Nipbl by siRNA do not change the gene expression of *HOX* and *PCDHB* genes. (a) *NIPBL* mRNA levels upon transfection with *NIPBL* siRNA. (b) *PCDHB* mRNA levels after transfection with siRNA *NIPBL*. (c) Gene expression levels of *HOX* and *HOX* related genes upon siRNA *NIPBL* transfection. (d) *NIPBL* protein levels upon siRNA *NIPBL* transfection compared to control cells transfected with negative control siRNA. Different amounts of loaded protein extract in the control are indicated for a better comparison.

Secondly, we generated a plasmid containing *NIPBL*-Patient 2 mimetic mutation. Upon nucleofection of control cells with control-*NIPBL* and mutated-*NIPBL*

plasmids, we found that the mutated-NIPBL plasmid is highly toxic for the cells and most of the cells died (see Figure below). As expected, the remaining survival cells did not express mutated-NIPBL as measured by RT-PCR (Figure Reviewer 2). This technical problem prevented us from doing the experiment.

Figure Reviewer 2. Transfection with NIPBL-Patient 2 mimetic mutation is toxic for the cells. NIPBL mRNA levels were quantified by RT-qPCR upon transfections with the indicated plasmids (left panel). Survival of transfected cells was observed 24h post-transfection (right panel).

We are currently trying to create a patient mimetic mutation by genome editing by CRISPR in control cell lines. However, the first experiments that we performed to obtain the genome editing in control fibroblasts were not successful. Although, this is something we are still trying to pursue in the lab, it might take us quite a long time to get it. In any case, we think that this is out of the scope of the manuscript at this point. The new data in Supplementary Figure 4 already allow us to study primary and secondary gene expression at PCDHB and HOX clusters, strengthening our conclusions.

- Although Nipbl levels are down in the HOXD cluster which is consistent with the reduced expression of HOXD genes.
- HOXB and C are hugely upregulated in patients but this is not discussed in the paper. Are they affected due to the altered expression of HOXD?

NIPBL chromatin binding levels measured by ChIP-seq are down in the HOX cluster, including HOXB, HOXC and HOXD clusters. Why *HOXD* genes are downregulated while *HOXB* and *HOXC* genes are upregulated in the patients is something we still do not understand. The HOXD cluster genome organization has been greatly studied and it is known to be located at the boundary between two TAD's. However, the HOXB and HOXC clusters have not been studied so well at the level of genome organization and not clearly defined TAD's are observed in HiC data. In our new data based on the intra-TAD prediction tool, we observed that the HOXC cluster is located upstream to one predicted intra-TAD in control cells, while for the

HOXB cluster the prediction tool does not predict a strong TAD (grey color indicated lower score TADs). By contrast, the HOXD cluster it is located at the boundaries between two TAD's (as described by the published HiC data). Therefore, our hypothesis is that differential chromatin organization around the HOX clusters might regulate gene expression in different ways.

Figure Reviewer 3. Representation of the intra-TAD prediction tool around the HOX clusters. The intra-TAD's predicted scores are depicted in grayscale (black is the higher TAD score while light grey will be the lowest score). The amount of represented TAD's (represented as horizontal lines) is also an indication of the likelihood of being an intra-TAD. The red squares mark the extend of the HOX cluster.

- Nipbl ChIPseq data was shown in patient 3, what about other patients?

Are the peaks that lost in patient 3 are consistent with other patient cells? The use of independent patient data will strengthen the conclusion drawn about the common genes that are dysregulated.

We thank the reviewer for this suggestion. We have validated some interested regions (including the HOXB, C and D (EVX2 is included in the HOXD cluster) among others) by NIPBL ChIP-qPCR in three new patients, patients P1, P2 and P4, suggesting a general trend in the reduction of NIPBL-chromatin association in the CdLS patients (new supplementary Figure 6).

Reviewer #2 (Remarks to the Author):

This manuscript by Garcia et al. investigates transcriptional and cohesin localisation changes in primary fibroblasts from patients suffering from Cornelia de Lange syndrome (CdLS), a developmental disorder that can affect many parts of the body and is associated with mutations of the cohesin loader factor NIPBL. The authors report that cohesin levels are not affected in CdLS patient-derived fibroblasts, however cohesin stability on chromosomes is reduced. Transcriptional analysis of these cells revealed alterations in developmental genes suggesting that the regulation of these genes might not be tightly regulated during development/differentiation in these individuals.

The authors then studied the distribution of NIPBL and cohesin on the genome of the CdLS fibroblasts by Chip-seq and reported a reduction in NIPBL sites at promoters. Cohesin was found to be increased at sites occupied by NIPBL but reduced at CTCF sites suggesting that cohesin traslocation is impaired. This is cosistent with recent data demonstrating a role for NIPBL stimulating cohesin ATPase activity and loop extrusion.

Overall, this study is well executed and contains a wealth of information that will be of great interest to scientists in the field, however I would suggest that the authors perform HiC analysis on CdLS fibroblasts and support the changes observed in their current ChiP-seq analysis with the expected alterations that this should cause in genome architecture in these cells. Besides the HiC analysis I don't have any major concerns that need to be addressed.

We thank the reviewer for the comments and suggestion. The reviewer suggested to perform Hi-C analysis. However, the normal Hi-C experiments do not have enough resolution to detect the small differences expected in the CdLS derived cells. High resolution Hi-C will be required (subkilobase resolution), however this is a pretty expensive approach completely out of our current funding. We are trying to increase our funding to perform high resolution Hi-C in the future or any other related (and cheaper) approach like capture HiC, Hi-ChIP or micro-HiC in collaboration with some experts in the field.

In the meantime, we took advantage of the *in silico* approach published in Matthews and Waxman (eLife 2018), which predicts intra-TADs based on ChIP-seq data of cohesin and CTCF. We have used and adapted the pipeline (consulting with the authors) from the original published paper to perform intra-TAD loop prediction on the control and CdLS-derived fibroblasts (new Figure 7 and supplementary Figure 8). In addition, validation of several predicted intra-TAD's was done by 3C-qPCR in control and CdLS derived-fibroblasts detecting a 30-40% reduction in the 3C interactions in two patients (Figure 7). These results suggest the presence of weaker chromatin interactions in the CdLS-derived fibroblasts.

Reviewer #3 (Remarks to the Author):

The manuscript by Garcia et al. presents a study on transcriptome analysis of CdLS patients' derived fibroblasts. It also reports analysis of genome-wide distribution and CpG islands. The topic is clearly interesting, and it goes beyond the chromatinopathies field as cohesins and their regulation are fundamental in cell biology.

Despite the clear importance of such studies, a number of inaccuracies must be addressed before possibly considering the work for publication.

First and foremost, the bulk of the study is based on primary fibroblasts passaged 2 to 6 times. This is very troubling as, especially considering the low number of samples, the same exact passage should have been used for comparison. Moreover, only 2 healthy donors have been used. This is not acceptable. No information on genetics of this healthy donors is reported.

The gene expression data was obtained using the lower possible passage (passage 3 since we received the fibroblasts mostly at passage 2). However, the following validation assays have been done within the next passages (4-8). We are sorry for the ambiguity in the materials and methods sections and we have explained this better in the new version of the manuscript.

As indicated in Table S1, the control cell lines are derived from 2 healthy controls, a boy and a girl of similar ages to the CdLS patients. Originally, the experiment was designed including 4 controls (as indicated in the nr-report submitted with the original manuscript). To obtain control samples from healthy children was quite complicated since they have to be related to the same Hospital and undergo some kind of surgery to be willing to provide the samples and the ethical consent. For this reason, in order to have more controls we included two adult controls (parents of CdLS Patient 4), since relatives of the CdLS patients are easier to recruit. Initial PCA and clustering analyses of the gene expression data showed a clear batch effect due to the age (samples >15 years old clustered together). For this reason,

initially we carried out the analysis using only the two controls children derived-fibroblasts in our original manuscript.

We agree with the reviewer that two controls are below optimal (and was not intended that way). We have now included the gene expression data including the 4 controls samples as originally designed. We have introduced a batch effect correction for the age after normalization of the data. Similar results are obtained and we also observed deregulation of the *HOX* and protocadherin genes using the 4 controls (new Figure 3). Just note, that we avoid to performed deep comparison on DEG's since we are aware that the final list of DEG's might vary depending on the controls used.

Further:

1-Western Blot analysis must be revised: quantification should be used for saying "similar" or "more". The methods do not explain if results presented have been obtained by one blot with 5-7 antibodies, by stripping 7-8 times or by single blotting with actin. In the first case, which could lead to cross-reaction, the full membrane should be presented. in the second case, quantifying a multiple-stripped membrane could lead to mistake. In the last case, each experiment should be presented with the specific control (at least in supplementary material). Finally, how many times were WB repeated? How many experimental and how many technical controls?

All the original data is now presented in the source data files containing all the quantifications. The triplicates presented in the source data are biological triplicates. Most of the antibodies are detected by cutting the membrane in different pieces (to detect different protein sizes) as can be observed in the source data for the western blots. We only stripped the membrane once, to detect SMC1A and RAD21 that have similar size.

The quantifications of the western blots of Figure 1a are now included in Figure supplementary 1a and more details can be seen in the source data. No significant differences were observed in the quantification and statistics analysis.

2-For immunofluorescence experiments: similar is not accurate. Again, this needs replicates, quantification and stats.

The quantifications of the immunofluorescence are now included in Figure supplementary 1b and more details can be seen in the source data. No significant differences were observed in the quantification and statistics analysis. Moreover, the detailed data from the biological triplicates can be seen in the source data file.

3-Photobleaching: was it performed on one patient vs on control? Only once? Replicates and stats are missing.

We apologize since we noticed that the figure was not interpreted correctly. We have rewritten the figure legend to make clear that two patients were used (denoted P2 and P3 in figure 2b and figure 2c). The representative image cells in figure 2a included only one patient, but the quantification in figure 2b and 2c included the two patients (P2 and P3), with replicates and stats. We have included also representative images for the second patient for clarification. Moreover, the detailed data is now included in the source data file.

4-The transcriptome analysis is very confusing: if fibroblasts were used, why developmental genes (including skeletal and urogenital) should be downregulated? Such pathways should not be different in the used cell lines. Moreover, how can author speculate in the results and discussion sections about implication of their findings during development and CdLS phenotype using differentiated primary cells? I believe that the modelling system does not support authors' interpretation.

We agree with the reviewer that gene expression changes in developmental genes are striking since we used CdLS-derived primary cells. However, we have validated the gene expression changes extensively with the same patients and new patients not included in the -omic gene expression assay (including two SMC1A-mutated patients, Supplementary Figure 4). In addition, the data included in the new Supplementary Figure 3, clearly showed that many gene expression changes can be rescued by re-introduction of wild type *NIPBL*. This demonstrated that at least some of the reported altered gene expression are direct, and not secondary effects, and strengthen our results.

We have removed the speculations regarding development and CdLS phenotype in the results section and tune down them in the discussion part. Our speculation tried to reconcile the fact that we see unexpected differences in developmental genes in primary fibroblast and is based on published data suggesting that TAD and loops are mostly constant among cell types, while they change during evolution or embryonic development as described in the last years.

5-A general lack of accuracy is noted: nomenclature of genes and proteins is wrong throughout the text. Please refer to HUGO guidelines. Methods are not detailed. A number of previously published papers on the topic and with similar experiments are missing. Please carefully assess the literature and bibliography. Figures are difficult to follow: please use panels, arrows and a general order in figure preparation. Finally, figure legends should explain everything in the figure, not just the main finding. Please revise figure legends thoroughly.

We thank the reviewer for the suggestions. We have now followed the HUGO guidelines for the nomenclature.

We have included more detailed materials and methods and incorporated the sentence that "more detailed methods and scripts can be obtain upon request to the corresponding authors".

We have assessed the literature and include missing papers and the more relevant newly published in the last months. We missed some publications due to limited space (we are quite over the recommended amount of cited papers).

We have rearranged the figures and revised the figure legends.

REVIEWERS' COMMENTS

Reviewer #1 (Remarks to the Author):

The authors have made significant improvements to the revised version of the manuscript. For example rescue with wild type NIPBL and also new Insilico analysis of chromatin structure followed by experimental validations.

The publication is an important contribution to understanding the possible role of NIPBL mutations in causing gene expression changes that contribute to the CdLS phenotype.

As the authors agree there is much more work is needed to clearly delineate the CdLS mechanism which is beyond the scope of this work and I believe this version is suitable for publication.

Best wishes
Pradeepa Madapura

Reviewer #2 (Remarks to the Author):

The revised manuscript has been improved with the addition of new data, despite the fact that the authors did not perform the HiC analyses suggested, they were able to address the concern raised through analysis of available datasets. I am therefore satisfied with their revisions and support publication of their work.

Reviewer #3 (Remarks to the Author):

The revised version of the manuscript has addressed major concerns. New data and revised methods are now better presented and support conclusions. Figures and figure legends are now of high standard. Nomenclature has improved but there are still some mistakes (ex. line 236 NIPBL should be in italics; line 254, mouse genes/protein have different nomenclature; line 281; etc....) please carefully revise. The last sentence of the discussion is very hypothetical and I recommend remove it (line 541-453) and modify title of the discussion section accordingly.

REVIEWERS' COMMENTS

Reviewer #1 (Remarks to the Author):

The authors have made significant improvements to the revised version of the manuscript. For example rescue with wild type NIPBL and also new Insilico analysis of chromatin structure followed by experimental validations.

The publication is an important contribution to understanding the possible role of NIPBL mutations in causing gene expression changes that contribute to the CdLS phenotype.

As the authors agree there is much more work is needed to clearly delineate the CdLS mechanism which is beyond the scope of this work and I believe this version is suitable for publication.

Best wishes

Pradeepa Madapura

We thank the reviewer for the comments.

Reviewer #2 (Remarks to the Author):

The revised manuscript has been improved with the addition of new data, despite the fact that the authors did not perform the HiC analyses suggested, they were able to address the concern raised through analysis of available datasets. I am therefore satisfied with their revisions and support publication of their work.

We thank the reviewer for the comments.

Reviewer #3 (Remarks to the Author):

The revised version of the manuscript has addressed major concerns. New data and revised methods are now better presented and support conclusions. Figures and figure legens are now of high standard.

We thank the reviewer for the comments.

Nomenclature has improved but there are still some mistakes (ex. line 236 NIPBL should be in italics; line 254, mouse genes/protein have different nomenclature; line 281; etc....) please carefully revise.

We thank the reviewer for pointing this. We have fixed some mistakes and the names of genes are now in italics. For other model organisms (mouse, zebrafish) we have now followed their nomenclature.

The last sentence of the discussion is very hypothetical and I recommend remove it (line 541-453) and modify title of the discussion section accordingly.

We have removed the last sentence of the discussion as recommended by the reviewer and the discussion subtitles following the Nature communications manuscript formatting guidelines.